# High-purity ethylene production via indirect carbon dioxide electrochemical reduction

Wenpeng Ni[1], Houjun Chen[1], Naizhuo Tang[1], Ting Hu[1], Wei Zhang[1], Yan Zhang[1] & Shiguo Zhang ⓘ [1] ✉

High-purity ethylene production from $CO_2$ electroreduction ($CO_2$RR) is a coveted, yet arduous feat because the product stream comprises a blend of unreacted $CO_2$, $H_2$, and other off-target $CO_2$ reduction products. Here we present an indirect reduction strategy for $CO_2$-to-ethylene conversion, one that employs 2-bromoethanol (Br-EO) as a mediator. Br-EO is initially generated from $CO_2$RR and subsequently undergoes reduction to ethylene without the need for energy-intensive separation steps. The optimized AC-Ag/C catalyst with Cl incorporation reduces the energy barrier of the debromination step during Br-EO reduction, and accelerates the mass-transfer process, delivering a 4-fold decrease of the relaxation time constant. Resultantly, AC-Ag/C achieved a $FE_{ethylene}$ of over $95.0 \pm 0.36\%$ at a low potential of −0.08 V versus reversible hydrogen electrode (RHE) in an H-type cell with 0.5 M KCl electrolyte, alongside a near 100% selectivity within the range of −0.38 to −0.58 V versus RHE. Through this indirect strategy, the average ethylene purity within 6-hour electrolysis was $98.00 \pm 1.45$ wt%, at −0.48 V (vs RHE) from the neutralized electrolyte after $CO_2$ reduction over the $Cu/Cu_2O$ catalyst in a flow-cell.

Ethylene serves as a crucial feedstock for the plastics industry, particularly for the production of polystyrene, polyvinyl chloride, and polyethylene, which find extensive use in packaging, automotive and electrical applications[1,2]. Presently, the mainstream route for manufacturing ethylene involves the steam cracking of hydrocarbons, such as ethane, propane, and naphtha at high temperatures (800 to 850 °C) over alumino-silicate catalysts[3]. This process is highly carbon-intensive and powered by non-renewable energy sources (e.g., natural gas or coal), releasing copious amounts of $CO_2$[4]. Alternativity, the electrochemical conversion of $CO_2$ to ethylene, driven by renewable electricity, offers a promising means of reducing carbon footprint and contributing to a circular carbon economy. But this technology is still beset by huddles like low efficiency and selectivity caused by the intricate reduction routes and 12-electron transfer process involved in $CO_2$-to-ethylene conversion. To tackle these issues, diverse regulation strategies have been developed for Cu-based catalysts, for stabilizing key intermediates (*CO and *CHO) and promoting C-C coupling. These strategies encompass phase and defect engineering[5–9], surface molecular modification[10–14], grain boundaries construction[15–21], heteroatom doping[22–25], oxidation state control[26–28], local pH modulation[29–31] and tandem catalyst design[32]. Encouragingly, Cu/polyamine electrodes have demonstrated Faradaic efficiency for ethylene ($FE_{ethylene}$) surpassing 85.0%[12].

Despite achieving remarkable selectivity towards ethylene, the gas product stream of $CO_2$RR still consists of a mixture of unreacted $CO_2$, $H_2$ (derived from water dissociation), and other off-target reduction products like CO and $CH_4$ (Fig. 1a). These impurities arise due to the limited single-pass conversion efficiency of $CO_2$ molecules and the occurrence of other competition reactions. Consequently, obtaining high-purity ethylene (e.g., polymer-grade ethylene streams), which is crucial for the durability of the Ziegler-Natta catalyst in polyethylene preparation[33], necessitates additional downstream separation processes, which is energy-intensive[34]. On the other hand, the electrolytic cell configuration design has proven to be effective in achieving high-purity $CO_2$RR products directly[35]. For example, using a flow-through electrochemical cell, an ethylene purity of 52 wt% was obtained without $CO_2$ in the product[36]. Regrettably, this still falls short of industrial requirements, and the removal of other inevitable $CO_2$ reduction products remains an insurmountable challenge. Therefore,

[1]College of Materials Science and Engineering, Hunan University, Changsha, China. ✉e-mail: zhangsg@hnu.edu.cn

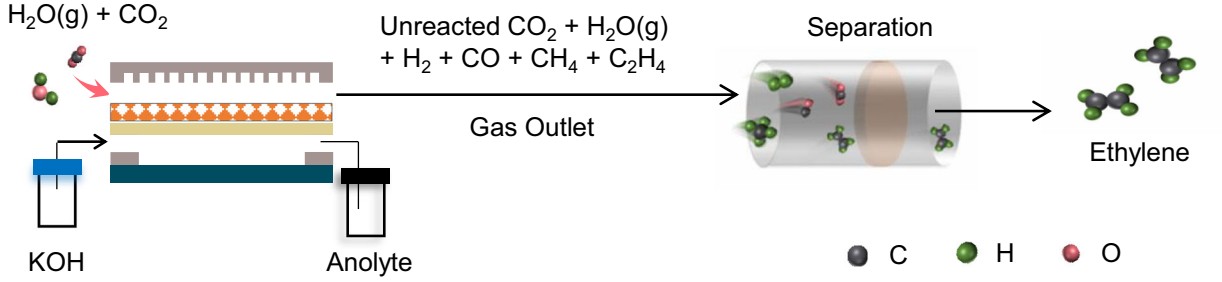

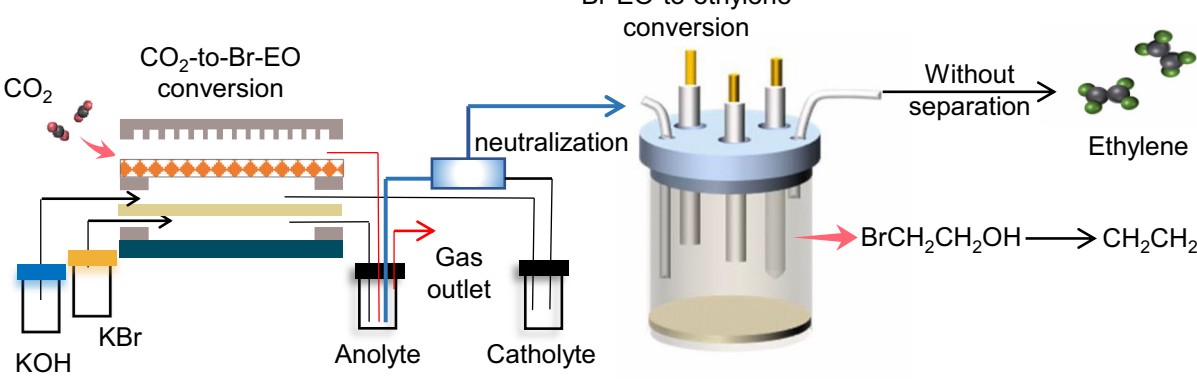

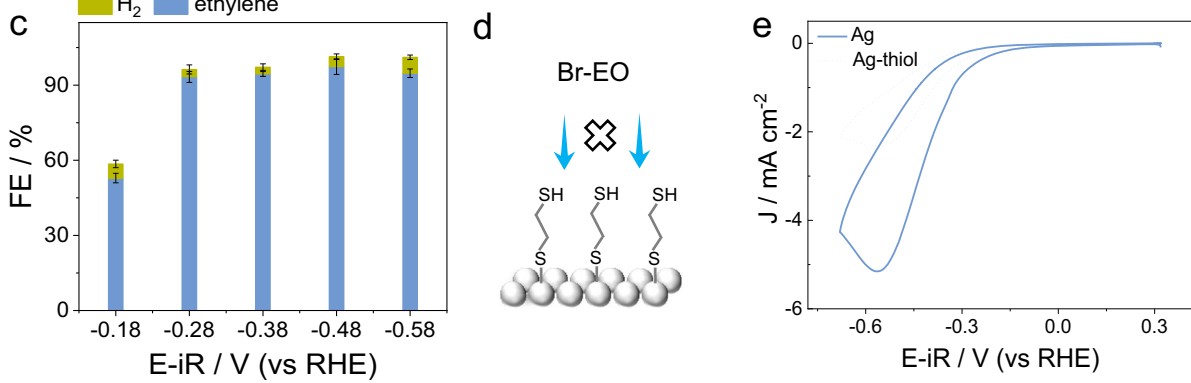

**Fig. 1 | Indirect electroreduction route design. a** Direct reduction, and (**b**) indirect reduction of $CO_2$ for high-purity ethylene production (2-bromoethanol: Br-EO). **c** $FE_{ethylene}$ and $FE_{H2}$ for electrochemical reduction of Br-EO over commercial Ag foil electrode ($1 \times 1\,cm^2$) in 0.5 M KCl electrolyte containing 50 mM Br-EO, under $CO_2$ bubbling (30 sccm). Error bars correspond to the Standard Deviation of three independent measurements. **d** Schematic illustration for the shielding effect of specific adsorbed thiol molecules (2-bromoethanol: Br-EO). **e** CV curves for Br-EO reduction (50 mM) in 0.5 M KCl electrolyte with and without thiol additive (50 μM). The scan rate is 10 mV s⁻¹.

a novel reaction paradigm that enables the direct production of high-purity ethylene without any separation step would be highly desirable.

In this work, we present an alternative strategy to produce high-purity ethylene through an indirect reduction process, wherein $CO_2$ is converted into 2-bromoethanol (Br-EO) first, which can be further reduced to ethylene even in the presence of other liquid $CO_2RR$ products. We demonstrated that an electrochemically activated Ag/C catalyst (AC-Ag/C) can achieve an unparalleled ~100% ethylene selectivity for Br-EO reduction, with a low onset potential of −0.08 V (vs RHE), thanks to Cl-incorporation and lower mass-transfer barrier. The mechanism of Br-EO reduction was explored by in-situ attenuated total reflectance-surface-enhanced infrared absorption spectroscopy (ATR-SEIRAS),

density functional theory (DFT) calculations, and electrokinetic analysis. Finally, through this indirect route, ethylene with the highest purity of 99.47 wt% was achieved from the electrolyte after $CO_2RR$ over $Cu/Cu_2O$, without any residue $CO_2$ and gaseous $CO_2$ reduction products. The average purity within 6-hour electrolysis was $98.00 \pm 1.45$ wt%, and the corresponding total electrical work was calculated to be 310.99 GJ/tonne ethylene, which is far lower than that of the direct $CO_2$-to-ethylene route.

## Results

### Indirect reduction strategy for ethylene production

Conventional $CO_2$ reduction mechanisms often yield intermediates that can diverge into different products, impeding precise control over

the reaction for high-purity ethylene production. Consequently, we advocate for a novel reaction paradigm, presenting an indirect pathway to circumvent the modulation of species and the strength of varied adsorbed intermediates in $CO_2$ reduction. This strategy requires a $CO_2$ reduction product to act as a mediator, which undergoes a further electrochemical reduction step to convert it into ethylene (Fig. 1b). This mediator should be able to be exclusively reduced to ethylene and have an onset potential that is more positive than that of all other co-generated liquid chemicals in the electrolyte. Meanwhile, the unreacted $CO_2$ in the tail gas can be directly circulated into the cathodic input to amalgamate with fresh $CO_2$ and is subsequently directed into the cathode chamber for further reduction. This approach allows surplus $CO_2$ gas to undergo multiple passes through the cathode, thereby avoiding excess $CO_2$ emission.

The typical liquid products of $CO_2$ electroreduction can be divided into three categories, namely carboxylic acid, aldehyde, and alcohol. The carboxylic acid can be transferred to $CH_4$ or $C_2H_2O_2$ via a decarboxylation or dehydration process, while the reduction of aldehyde has been reported to give the alcohol as the preferential product. Only alcohol was shown that olefin can be yielded from a reduction reaction, but possesses a high kinetic energy barrier due to the activation of the inert C-H bond. Therefore, our focus was turned to searching for alcohol with a replacement for the C-H bond as the mediator. The C-X bond (X = Cl, Br, I) was thought more active than its C-H counterpart, and the electrochemical breakage of the C-X bond has been realized[37,38]. Therefore, the halogen-containing alcohol can be one potential candidate. Interestingly, 2-bromoethanol (Br-EO), bearing a C-Br bond, was found generated from $CO_2$ electroreduction in a $Br^-$-containing electrolyte or from the reaction between ethylene, derived from $CO_2$ reduction, and $Br_2$ from the anodic oxidation of $Br^-$[39–41].

For the initial assessment of the conversion of Br-EO to ethylene, we utilized polycrystalline Ag-foil as a catalyst, owing to its high efficiency in reductively cleaving the C-halide bond[42,43]. By employing a 0.5 M KCl aqueous electrolyte (Fig. S1), we observed a positive shift in the potential required to achieve a current density (J) of $1\,mA\,cm^{-2}$ ($E_{J=1\,mA\,cm^{-2}}$) in linear scanning voltammetry (LSV) curve from −0.72 to −0.19 V (vs RHE) upon addition of 50 mM Br-EO (Fig. S2). Online GC spectra and $^1H$ NMR revealed that the sole reduction product of Br-EO was ethylene (Fig. S3), along with a minor amount of $H_2$ from water splitting. Even at −0.18 V (vs RHE), the detectable ethylene was formed (Fig. 1c). As the applied bias was negatively shifted, the $FE_{ethylene}$ gradually increased, and the highest selectivity of 97.7 ± 3.1% was achieved at −0.48 V (vs RHE). These results strongly demonstrate that highly selective formation of ethylene can occur over Ag foil via Br-EO reduction, even though the highest production rate of ethylene was only $0.22\,mmol\,cm^{-2}\,s^{-1}$. Furthermore, to distinguish between inner- and outer-sphere reaction routes, we employed short-chain thiol as a probe, which hinders the adsorption of reactant while not affecting the electron tunneling from the electrode to the solution species (Fig. 1d)[44]. As depicted in the cyclic voltammetry (CV) curves (Fig. 1e), in the electrolyte containing 50 μM 1,2-ethanedithiol, the peak current for Br-EO reduction decreased to $-2.33\,mA\,cm^{-2}$, which is only half of the original value. Therefore, we can infer that Br-EO is reduced through an inner-sphere reaction mechanism, indicating the selectivity and kinetic rate for Br-EO reduction are associated with the microstructure of the electrocatalyst. Consequently, it can be postulated that the production rate of ethylene can be further boosted by catalyst design.

## Reduction of Br-EO over electrochemically activated Ag electrode

To enhance the efficiency of Br-EO reduction, we have devised two additional Ag-based catalysts through heteroatom doping and nanostructural optimization. The first one was synthesized by electrochemically activating Ag foil in KCl solution using five CV scanning cycles ranging from 0.05 to 1.05 V (vs RHE, denoted as AC-Ag). Once the oxidation potential was reached, the Ag electrode underwent a noticeable color change from its original silver-bright hue to a dark black, with AgCl being the predominant component (Figs. S4 and S5). The electrode subsequently reverted to a pale-yellow color during the reductive scanning. Unlike the rough surface texture with visible scrape marks (Fig. S6), the scanning electron microscope (SEM) image of AC-Ag displays a nanoporous morphology, comprising worm-like particles and some isolated particles that dispersed over the surface, ranging in size from 300–700 nm (Fig. 2a and S7). Cross-sectional observation showed a new activation layer with a thickness of 2.5 μm (Fig. 2b and S8). Furthermore, high-resolution TEM (HRTEM) imaging of the ultrasonic exfoliated Ag activation particles exhibited a well-defined crystalline structure (Fig. 2c and S9), with the Ag(111) face being clearly delineated in the enlarged image (Fig. 2d). The selected-area diffraction pattern was aligned with the Ag[−1,−1,0] zone, displaying spots indexed to the (1,−1,1), (2,−2,0), and (0,0,2) planes (Fig. 2e).

The second Ag catalyst is the electrochemically activated Ag/C (AC-Ag/C), which exhibits nanoparticles dispersed throughout the carbon matrix with a mean size of 10.5 ± 0.7 nm (Fig. 2f). The Ag content was determined to be 22.9 wt% by inductively coupled plasma optical emission spectrometry. To dissect the lattice structure of the Ag nanoparticles, the rich facets in the HRTEM image were analyzed by fast Fourier inverse transform (IFFT) (Fig. 2g). The integrated pixel intensity corresponding to the direction marked in the blue and orange box of Fig. 2g, gave the interplanar spacings of 0.24 and 0.20 nm, respectively, which are assigned to the (111) and (200) planes of Ag (Fig. 2e). The peak positions in X-ray diffractometer (XRD) patterns of all three Ag-based catalysts are all well consistent with the pure Ag crystalline phase with a cubic structure (Fig. 2i, space group Fm-3m, JCPDS No. 04-0783, a = b = c = 4.0862 Å). The primary crystalline origination of pristine Ag foil is (220) and (200), while AC-Ag and AC-Ag/C are (111).

The Ag $3d$ X-ray photoelectron spectroscopy presented in Fig. 2j indicates that electrochemical activation upshifts the peaks of $3d_{5/2}$ and $3d_{3/2}$, suggesting an increase in the Ag oxidation state. Further inspection verifies that this is mainly due to the incorporation of Cl atoms, as evidenced by the doublets of Cl $2p_{3/2}$ (198.98 eV) and $2p_{1/2}$ (197.35 eV) for the Ag-Cl interaction (Fig. 2k)[45]. Notably, variations of the binding energy for Ag $3d$ peaks in AC-Ag/C are larger than that of AC-Ag, which is likely caused by the higher doping content of Cl in Ag/C (the Cl/Ag atomic ratios are 0.20 and 0.34 for AC-Ag and AC-Ag/C). Based on these two samples, the influence of Cl-doping and nanostructure on the activity of Ag catalysts for Br-EO-to-ethylene conversion can be evaluated.

The activity of Br-EO reduction over different Ag catalysts was initially evaluated by LSV testing. To simulate the electrolyte microenvironment after $CO_2RR$, a KCl solution containing 50 mM Br-EO was saturated by $CO_2$. Both AC-Ag and AC-Ag/C showed a more positive onset potential and higher J than pristine Ag foil (Fig. S10). $E_{J=1\,mA\,cm^{-2}}$ was found to be −0.19, −0.07, and 0.03 V (vs RHE) for Ag foil, AC-Ag, and AC-Ag/C, respectively. The potentiostatic electrolysis gave an identical conclusion, with AC-Ag/C exhibiting the best activity. Of note, AC-Ag had a limiting current density ($J_d$) of $-34.0\,mA\,cm^{-2}$ at −0.38 V (vs RHE, Fig. 3a), whereas the largest J reached $-84.2\,mA\,cm^{-2}$ at −0.58 V for AC-Ag/C. Regarding the selectivity, $FE_{ethylene}$ higher than 95.0% was observed for both AC-Ag and AC-Ag/C at all studied potentials (−0.08 – −0.58 V vs RHE, Fig. 3b). Particularly, near-unity conversion of Br-EO to ethylene was obtained from −0.38 to −0.58 V (vs RHE, Fig. S11). Ag has also shown significant efficacy in the $CO_2$-to-CO conversion as previously reported[46,47], but no discernible formation of CO was observed in our present investigation due to the carefully chosen reduction potentials and the marked predilection for

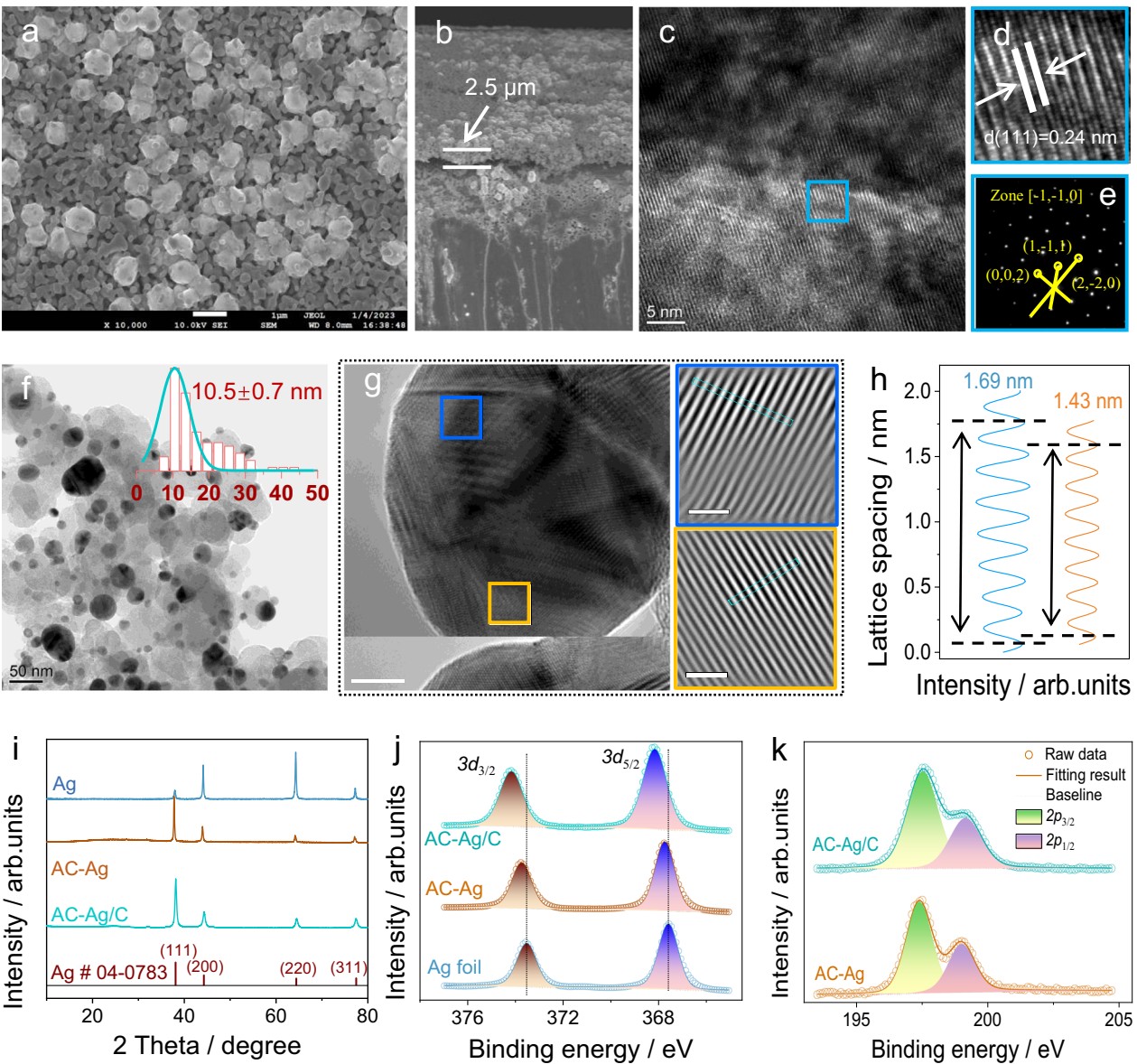

**Fig. 2 | Characterizations for different Ag-based catalysts. a** SEM image, and (**b**) cross-sectional SEM image of AC-Ag. **c** High-resolution TEM image, and (**d**) corresponding area zoom in the turquoise box of AC-Ag. **e** SEAD image of AC-Ag. **f** TEM, inset is the size distribution pattern, (**g**) High-resolution TEM images for AC-Ag/C, and selected IFFT patterns for areas in blue and orange box. **h** Integrated pixel intensity of AC-Ag/C in the selected blue and orange box in (**g**). **i** XRD patterns, (**j**) Ag *3d* XPS spectra of Ag, AC-Ag, and AC-Ag/C catalysts. **k** Cl *2p* XPS spectrum of AC-Ag and AC-Ag/C.

reduction of Br-EO (Fig. S12). The ethylene production rate of AC-Ag/C was estimated to be 1.57 mmol cm$^{-2}$ h$^{-1}$ at −0.58 V (vs RHE), which represents a 7.1-fold improvement over pristine Ag foil. The excellent activity of AC-Ag/C can be attributed to the electrochemical active Ag particles (Figs. S13–S14). Meanwhile, the observed high selectivity for Br-EO reduction is not correlated with the bubbled gas to the electrolyte, as confirmed by the identical FE$_{ethylene}$ between CO$_2$- and N$_2$-saturated electrolyte systems (Fig. S15). Subsequently, we screened the ethylene selectivity by decreasing the Br-EO concentration from 50 mM to 2 mM. In electrolytes with low concentrations, diffusion became the controlling factor under the higher overpotentials (Fig. S16). However, the high FE$_{ethylene}$ (> 95.0%) was well maintained at all potentials for electrolytes with Br-EO concentrations of 50−5 mM (Fig. 3c). Interestingly, even for a low concentration of 2 mM, such high selectivity was achieved at −0.18 and −0.28 V (vs RHE, Table S1). Using a flow-through cell, the ethylene selectivity exceeded 98.0% when

50 mM Br-EO was added, and the J surpassed those achieved in the H-type cell (e.g, −123.0 mA cm$^{-2}$ at −0.6 V, Fig. S17).

Upon the consumption of Br-EO, J gradually declined and FE$_{ethylene}$ dropped to below 90.0% after 5.5 h of testing at −0.38 V (vs RHE) over the AC-Ag/C catalyst (Fig. 3d). However, with the injection of 100 μL fresh Br-EO, both J and selectivity can be restored. Even after four cycles, there is no obvious attenuation of ethylene selectivity, indicating the robust stability of the AC-Ag/C catalyst for Br-EO reduction.

To assess the specific activity of different catalysts, electrochemical active surface area (ECSA), measured by Pb underpotential deposition (Fig. S18), normalized partial current density of ethylene was compared (J$_{ethylene/ECSA}$). AC-Ag/C still exhibits a higher J$_{ethylene/ECSA}$ than AC-Ag and Ag foil, indicating its superior intrinsic activity (Fig. S19). Moreover, the apparent activation energy was calculated according to the Arrhenius equation (Fig. 3e and S20). AC-Ag/C

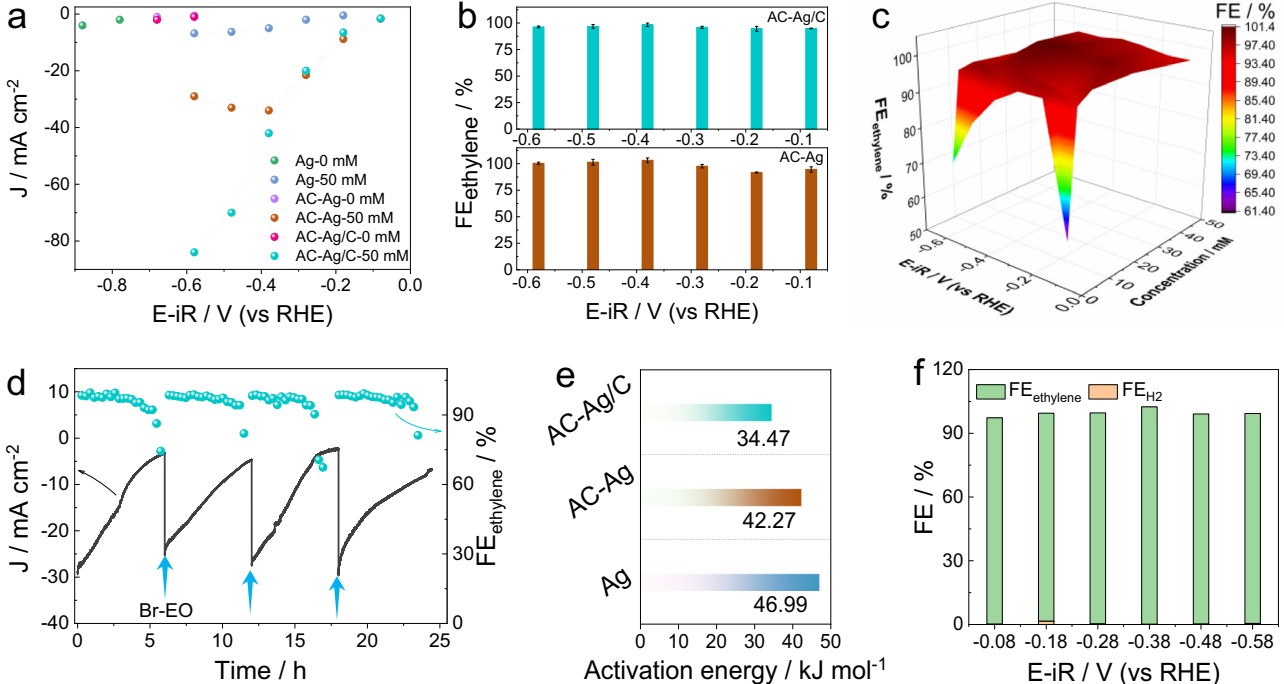

**Fig. 3 | Electrochemical performances. a** J of Ag, AC-Ag, and AC-Ag/C electrodes in 0.5 M KCl with 0 mM and 50 mM Br-EO. **b** FE$_{ethylene}$ of AC-Ag and AC-Ag/C catalysts in 0.5 M KCl with 50 mM Br-EO. Error bars correspond to the Standard Deviation of three independent measurements. **c** FE$_{ethylene}$ of AC-Ag/C in 0.5 M KCl containing Br-EO with different concentrations (2, 5, 10, 25, and 50 mM). **d** Stability of AC-Ag/C catalyst for Br-EO reduction. The arrow indicates the introduction of fresh Br-EO (100 μL). **e** The activation energy for Ag, AC-Ag, and AC-Ag/C. **f** FE of H$_2$ and ethylene of AC-Ag/C electrode in 0.5 M KBr electrolyte. The size of Ag and AC-Ag electrode is 1×1 cm². The mass loading of AC-Ag/C is 1 mg cm⁻².

possesses an activation energy of 34.47 KJ mol⁻¹, which is lower than that for AC-Ag (42.27 KJ mol⁻¹) and Ag foil (46.99 KJ mol⁻¹).

Furthermore, it should be highlighted that the identity of halogen ions does not affect the high ethylene selectivity. As shown in Fig. 3f, the FE$_{ethylene}$ surpasses 97.0% for the AC-Ag/C electrode across the potential range from −0.08 to −0.58 V (vs RHE) in the 0.5 M KBr aqueous solution.

## Discussion

### Mechanisms for the reduction of Br-EO over Ag-based catalysts

The mechanism for Br-EO reduction was explored by combining electrokinetic testing, in situ characterization techniques, and theoretical calculations. Initially, we measured the dependence of J$_{ethylene}$ on the concentration of different electrolyte components, namely Br-EO, H$_2$O, and KCl. First-order kinetics was found for the concentration of Br-EO at different potentials (Fig. 4a). The relationship between J$_{ethylene}$ and the concentration of H$_2$O was evaluated in a DMSO-based electrolyte with H$_2$O as the exclusive proton source. Intriguingly, we detected the formation of ethylene with a selectivity close to 100.0%, even in the absence of added H$_2$O (Fig. S21). With the addition of H$_2$O, there are no obvious changes in J and selectivity as displayed in Fig. 4b. This implies that the reduction of Br-EO on the Ag catalyst does not involve a proton-transfer step. Furthermore, t-BuOH, a quenching agent for proton[48], was added with a concentration of 50 mM, identical to Br-EO. Again, the FE$_{ethylene}$ and J kept stable after introducing t-BuOH (Fig. 4c). As for KCl, a zero-order relationship was acquired (Fig. S22). Therefore, the reduction of Br-EO is only governed by the transformation of Br-EO molecule.

Next, the adsorption/desorption behaviors of reactant and intermediates were monitored by in situ ATR-SEIRAS, from a potential range of 0.15 to −1.05 V (vs RHE) in D$_2$O-based KCl electrolyte (Fig. 4d). There are five dynamically changed peaks appeared. The peaks located at 1550 and 1400 cm⁻¹ can be ascribed to the -CH$_2$ and C-OH species in 2-bromoethanol. Meanwhile, the peak for the vibration of =CH$_2$ appeared at 1000 cm⁻¹, coupled with a new O-H vibration at 1635 cm⁻¹. As for the wide peak around 1200 cm⁻¹, it should be caused by the potential-dependent vibration of the Si prism. The dynamic changes in these peaks were visualized through the corresponding contour image (Fig. 4e). The peaks for Br-EO, including -CH$_2$ and C-OH peaks, exhibit positivity at high potentials, signifying the adsorption of Br-EO. However, these signals gradually diminish with the negative shift of applied potentials, ultimately culminating in the observation of negative peaks. This phenomenon suggests the fast transformation of adsorbed Br-EO. Concurrently, the intensified =CH$_2$ signal emerged at −0.05 V (vs RHE), which was close to the onset potential collected by electrochemical testing (−0.08 V vs RHE). This could be rationalized by the formation of ethylene. As for the great growth of the O-H peak, it should be derived from the adsorbed *OH species that were liberated from Br-EO. After the desorption of the -OH group, the OH⁻ ion is released into the electrolyte. To monitor the interfacial pH variation, we added a phenolphthalein indicator to the electrolyte. The color of the electrolyte near the working electrode turned pink when a bias of −0.28 V (vs RHE) was set (Fig. 4f), giving the local alkaline environment near the electrode. The halogen ions in electrolyte after Br-EO reduction were further analyzed by ion chromatography. In addition to the Cl⁻ from pristine KCl, a peak at the retention time of 7.48 min was observed (Fig. 4g), attributable to Br⁻. Taken together, a plausible mechanism of Br-EO reduction can be schematically illustrated in Fig. 4h. The breakage of Br-CH$_2$- leaves a Br⁻ ion, and subsequently, the -OH is removed. Then an intermediate of *CH$_2$CH$_2$* adsorbs onto the electrode surface. Finally, ethylene is produced after the desorption of *CH$_2$CH$_2$*.

Drawing on the established reaction pathway, we further shed light on the observed reaction trends for Ag foil, AC-Ag, and AC-Ag/C from an energetic perspective through DFT calculations. Considering the presence of Cl-dopants in the AC-Ag and AC-Ag/C electrode after

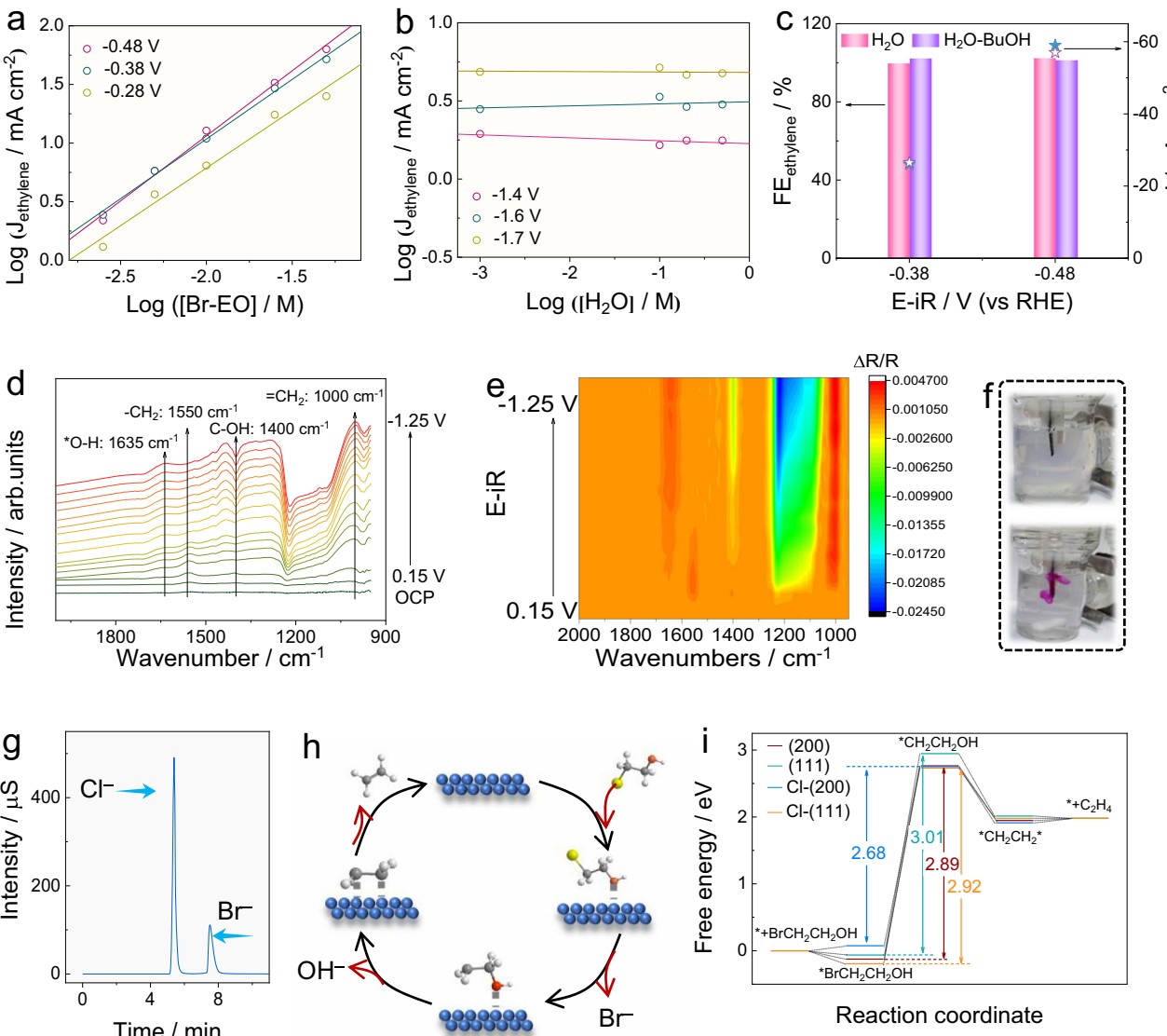

**Fig. 4 | Mechanism analysis for Br-EO reduction.** Dependence of $J_{ethylene}$ on the concentration of (**a**) Br-EO and (**b**) $H_2O$. **c** $FE_{ethylene}$ and J for AC-Ag/C electrode in the 0.5 M KCl containing 50 mM Br-EO at −0.38 and −0.48 V (vs RHE), before and after introducing 50 mM t-BuOH. The mass loading of AC-Ag/C is 1 mg cm⁻². **d** In situ ATR-SEIRAS for Br-EO reduction at different potentials collected in $D_2O$-based 0.5 M KCl electrolyte, and (**e**) its corresponding contour image. ΔR/R indicates the intensity variation related to the peak observed under OCP. **f** Digital photo of AC-Ag/C electrode in the electrolyte containing phenolphthalein indicator under OCP (up) and −0.28 V (vs RHE, down). **g** Ion chromatography of the 0.5 M KCl electrolyte collected after 2 h electrochemical testing at −0.48 V (vs RHE) over AC-Ag/C. **h** Schematic diagram of the reduction mechanism of Br-EO. Blue: Ag, Black: C, White: H, Red: O, Yellow: Br. **i** Free energy changes of different intermediates for Br-EO reduction. All potentials are IR compensated.

electrolysis (Fig. S23), four models based on the Ag (111) and Ag (200) planes were built, along with their Cl-incorporated counterparts (Fig. S24). The free energy changes (ΔG) for Br-EO adsorption, the breakage of the C-Br bond, the removal of -OH, and the desorption of *CH₂CH₂* were compared (Figs. S25−S27). Our results indicate that the adsorptions of Br-EO at Ag (111), Ag (200), and Cl-(111) are both spontaneous processes, while for Cl-(200) the adsorption energy barrier is 0.07 eV. Subsequently, the debromination step incurs the largest ΔG for all four models, meaning the formation of *OHCH₂CH₂ serves as the rate-determining step (RDS). Then both the removal of -OH and the desorption of *CH₂CH₂* are thermodynamically favorable. Notably, the ΔG of the RDS decreased after the Cl incorporation for (111) and (200) planes. The adsorption configuration showed that *OHCH₂CH₂ binds with Ag by Ag-O interaction (Fig. S26). In situ Raman spectra collected over AC-Ag/C catalyst also illustrated distinct peaks for Ag-O interactions (600−700 cm⁻¹)[49], of which are absent for that obtained in

electrolyte without Br-EO (Fig. S29). Interestingly, the Ag-O bond length of Cl-incorporated models is smaller than the pristine counterpart, suggesting the strengthened adsorption of *OHCH₂CH₂. For example, the Ag-O bond decreased from 3.049 Å in Ag (111) plane to 2.545 Å in Cl-(111) (Fig. S28). Considering the higher oxidation state of Ag for Cl-incorporated catalysts, the locally enriched positive charges enhance the electrophilicity of the adjacent Ag atom. Due to the O atom in Br-EO possessing a negative charge accumulation, the electrophilic nature of Cl-incorporated Ag site improves its oxygen affinity (Fig. S30), leading to the strong adsorption of the O-bound intermediate of *OHCH₂CH₂, thereby promoting the debromination process.

However, the superior activity of AC-Ag/C to AC-Ag cannot be interpreted by the lattice plane-dependent energetic differences since they possess similar crystalline structures. As per the Tafel analysis (Fig. 5a), the Tafel slopes for Ag foil, AC-Ag, and AC-Ag/C surpass

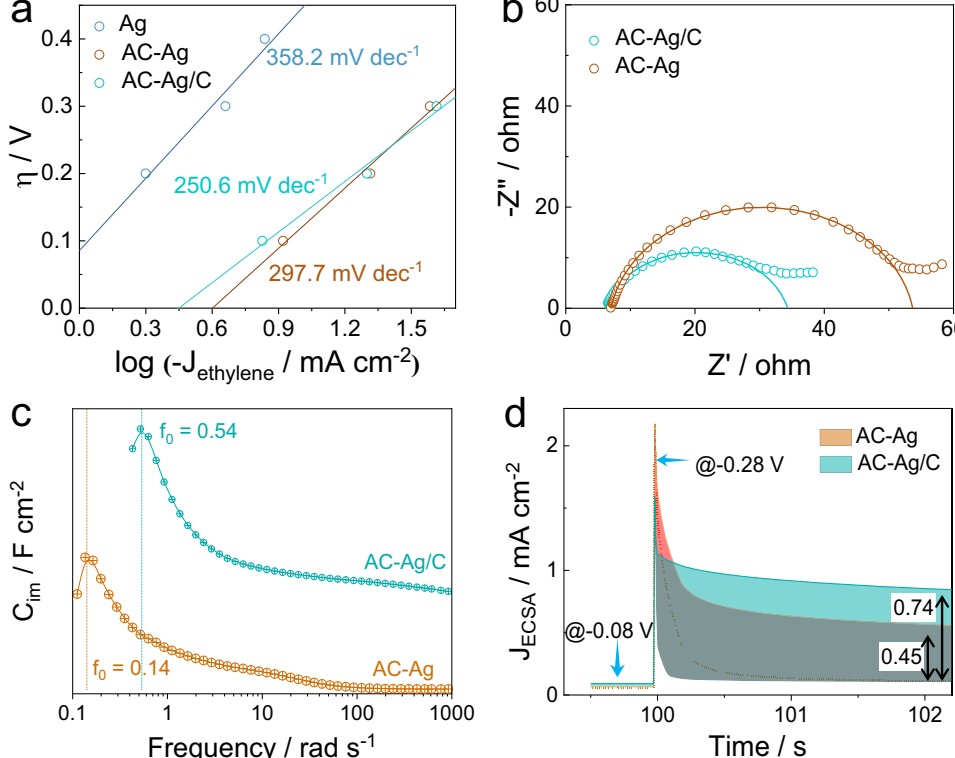

**Fig. 5 | Analysis of the influences of diffusions on the activity of AC-Ag and AC-Ag/C. a** Tafel plots for Br-EO reduction over Ag foil, AC-Ag, and AC-Ag/C. **b** Nyquist plots, **c** the imaginary part of complex capacitance, **d** transient kinetics in 0.5 KCl with and without Br-EO (20 mM) via pulse amperometry from −0.08 to −0.28 V (vs RHE), for AC-Ag and AC-Ag/C. All potentials are IR compensated. The size of Ag foil and AC-Ag is 1 cm². The mass loading of AC-Ag/C is 1 mg cm⁻².

118 mV dec⁻¹, implying that the reaction is controlled not only by electron transfer kinetics but also by the mass transfer process[50]. The LSV curves were collected under different stirring speeds, and the J increased with the improvement in magnetic stirring speeds from 0 rpm to 200 rpm (Fig. S31). These outcomes illustrate that the reduction of Br-EO is under mixed control of charge transfer and mass diffusion. For electron-transfer kinetic, electrochemical impedance spectroscopy (EIS) revealed that AC-Ag/C (28.24 Ω) exhibits lower interfacial transfer resistance ($R_{ct}$) than that of AC-Ag (46.67 Ω, Fig. 5b). This may be attributed to the larger interfacial electric field of AC-Ag/C (Fig. S32). Attention was then focused on the mass transfer kinetics, which can be assessed by the relaxation time constant ($\tau_0$). A shorter $\tau_0$ means a facilitated diffusion process[51]. We first analyzed the imaginary part of complex capacitance as a function of frequency based on the EIS response (Fig. 5c). The peak frequency ($f_0$) for AC-Ag and AC-Ag/C were 0.14 and 0.54, respectively. Hence, the calculated $\tau_0$ for AC-Ag/C, using the equation $\tau_0 = (2\pi f_0)^{-1}$, was about 0.29 s, which is only about one-quarter that of AC-Ag (1.12 s). Therefore, the mass transfer of Br-EO is significantly accelerated in the AC-Ag/C catalyst, which can lead to a larger response current. To demonstrate this, transient kinetics analysis was performed using the pulse amperometric technique[52]. Br-EO was pre-adsorbed at −0.08 V (vs RHE) for 100 s, followed by the application of a pulsed potential of −0.28 V (vs RHE). Eventually, a larger reductive current jump of 2.19 mA cm⁻² (normalized by ECSA) was recorded for AC-Ag/C, whereas this value was only 1.58 mA cm⁻² for AC-Ag (Fig. 5d). To sum up, the higher activity of AC-Ag/C than AC-Ag can be attributed to the faster interfacial electron transfer and rapid mass transfer kinetics.

### Indirect CO₂ reduction for high-purity ethylene production
It was reported that Br-EO can be generated via the reaction between ethylene (derived from CO₂RR) and bromine (from anodic oxidation)[39,53]. Electrocatalysts that promote CO₂-to-ethylene conversion will improve the selectivity for Br-EO. Herein we prepared a Cu/Cu₂O catalyst for this purpose. The PXRD analysis confirmed the mixed phase of Cu and Cu₂O (Fig. S33), and the TEM images revealed that the nanoparticles are interconnected with each other (Fig. S34). Furthermore, the HRTEM showed the highly crystalline structure of Cu/Cu₂O, in addition to the amorphous surface oxidation layer. The crystalline zone of Cu(111) (I), Cu₂O(111) (II), and Cu₂O(211) could be well distinguished (Fig. 6a). XPS evidenced the presence of Cu⁺ based on the distinctive peak around 917.3 eV in the Cu LMM Auger spectrum and the peak for lattice oxygen of Cu₂O in the O 2 s spectrum (Fig. 6b and S35–S36)[54,55].

The CO₂RR activity of this Cu/Cu₂O catalyst was evaluated by a cation-exchange membrane separated flow-cell with the catholyte of 1 M KOH and the anolyte of 1 M KBr, and the outlet gas of cathode flowed through the anolyte jar to convert residuum ethylene in the gas products to Br-EO. As shown in Fig. 6c, besides Br-EO, ethanol, aldehyde, acetate, ethylene, formate, CO, and H₂ were detected. The Br-EO Faradaic efficiency of 45.5% and 46.1% were obtained at −0.89 and −0.95 V, respectively, along with the partial current density of −97.8 and −138.8 mA cm⁻².

Next, the reduction sequence of all chemicals in electrolytes after CO₂RR was acquired by comparing the $E_{j=1\ mA\ cm^{-2}}$ in the LSV curves. Specifically, ethanol, acetate, and aldehyde possess the $E_{j=1\ mA\ cm^{-2}}$ comparable to that of the pristine electrolyte (Fig. 6d and S37), and their reduction potentials (−0.633 ~ −0.655 V, vs RHE) are considerably more negative than Br-EO (−0.060 V, vs RHE). This substantial potential difference indicates that the exclusive reduction of Br-EO is possible for the electrolysis of electrolytes after CO₂RR testing. Following continuous testing of CO₂RR over Cu/Cu₂O for 2 hours at −0.95 V (vs RHE), the remaining electrolyte was used for further electroreduction by the AC-Ag/C electrode at −0.48 V (vs RHE). Ethylene was observed

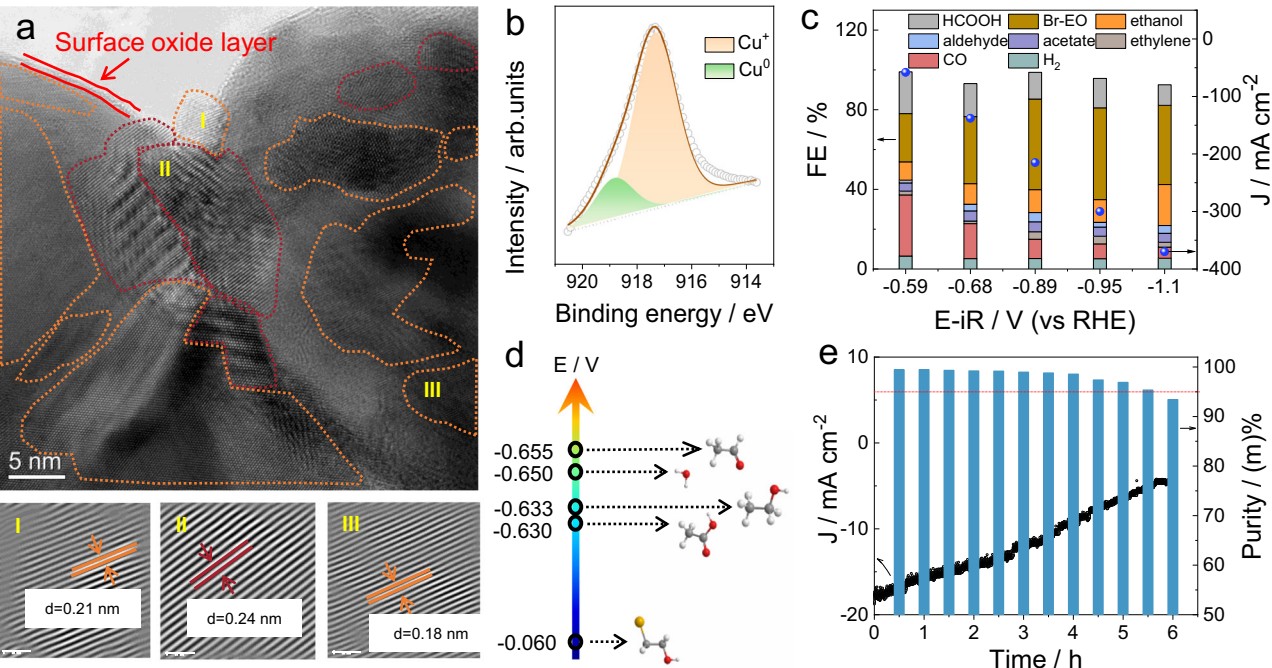

**Fig. 6 | Characterizations of Cu/Cu₂O and the ethylene production by indirect CO₂ reduction. a** High-resolution TEM image and selected IFFT patterns in I, II, III zone of Cu/Cu₂O catalyst. The scale bar for IFFT patterns is 1 nm. **b** Fitting result of Cu LMM Auger spectrum of Cu/Cu₂O catalyst. **c** FE and $J_{total}$ of CO₂ reduction products at different potentials over Cu/Cu₂O catalyst (1 mg cm⁻²), using H-type flow cell with the catholyte of 1 M KOH and the anolyte of 1 M KBr. **d** Comparisons of the initial reduction potentials for different C₂ products and water over AC-Ag/C catalyst. Black: C, White: H, Red: O, Yellow: Br. **e** J and the corresponding purity of ethylene collected by electrolysis in the neutralized anolyte of CO₂ reduction electrolytic cell. AC-Ag/C is the working electrode with mass loading of 1 mg cm⁻².

as the prominent product in the GC spectra (Fig. S38), with the FE_ethylene ranging from 98.4% to 95.2% in the first two hours, then gradually reduced due to the consumption of Br-EO in the electrolyte (Fig. S39). The largest ethylene purity is larger than 99.0 wt% (Fig. 6e). Within the continuous electrolysis of 6 h, the average ethylene purity is 98.00 ± 1.45 wt%. Such low potential for high-purity ethylene production has never been realized by direct reduction of CO₂ (Table S1).

According to the detected cell potentials for different current densities, we plotted the FE_Br-EO versus total current when a W_total of 564.3 GJ/tonne, equivalent to the electrical work for the direct CO₂-to-ethylene route[34], was accepted (Fig. S41, Supplementary Note 7). It can be found the required FE_Br-EO is lower than 25.76% under the current density of 5–100 mA cm⁻². Furthermore, the estimated total electrical work for our indirect route is 310.99 GJ/tonne of ethylene at the current density of 100 mA cm⁻² (Fig. S42, Supplementary Note 8), which is lower than that of the direct route.

In summary, we have presented an indirect electroreduction methodology to transform CO₂ into high-purity ethylene, utilizing Br-EO as a mediator, which can be reduced to ethylene over Ag-based catalysts via an inner-sphere reaction route. The optimized catalyst, Cl-incorporated Ag/C (AC-Ag/C), achieved FE_ethylene exceeding 95.0% even at a low potential of −0.08 V (vs RHE), along with a near-unity conversion from −0.38 to −0.58 V (vs RHE), with an ethylene production rate of 1.57 mmol cm⁻² h⁻¹ at −0.58 V (vs RHE). A detailed mechanism analysis disclosed that the reduction of Br-EO proceeds through the breakage of C-Br bond and subsequent the liberation of OH⁻ and desorption of ethylene. This supposed reaction pathway exhibited a first-order kinetics relationship between J_ethylene and the concentration of Br-EO and did not involve proton transfer. Micro-kinetic analysis suggested that Br-EO-to-ethylene conversion is under mixed control of charge transfer and mass diffusion. The Cl dopant has been found to increase the oxidation state of the Ag site, leading to enhanced oxygen affinity. This strengthens the adsorption ability for *OHCH₂CH₂, thereby promoting the debromination process which is

the RDS for Br-EO reduction. Moreover, nanostructural optimization for the Ag catalyst intensifies the mass-transfer kinetics, as evidenced by the shorter relaxation time constant and larger pulse current jump. Finally, based on this indirect route, we successfully produced high-purity ethylene with an average purity of 98.00 ± 1.45 wt% at −0.48 V (vs RHE) using the electrolyte after CO₂RR over Cu/Cu₂O catalyst, which showed the highest Br-EO selectivity of 46.1%. This contribution brings new insight into the design of CO₂ electroreduction technologies.

## Methods

### Materials preparation

Pristine Ag foil electrode: The Ag foil (>99.99%, Wuhu Yanjiao Co., LTD) was pretreated by 0.01 M hydrochloric acid, acetone, ethanol, deionized water (DI, 18.2 MΩ) obtained from an ultra-pure purification system (Master-S15Q, Hitech Instruments Co. Ltd., Shanghai, China), and then dried with a stream of nitrogen. The treated Ag foil was cut into a square with a geometric area of 1 cm², which was used as the pristine Ag foil electrode.

AC-Ag: The Ag foil electrode was activated in 0.5 M KCl aqueous solution by cyclic voltammetry within the potential window of 0.05 V to 1.05 V (vs. RHE). The scan rate is 10 mV s⁻¹. After five cycles, a pale-yellow electrode was obtained, which is denoted as AC-Ag. The in situ activated AC-Ag was used as a working electrode for Br-EO reduction without any treatment.

AC-Ag/C catalyst: Ag/C catalyst was prepared by wet chemical method. Silver nitrate (AgNO₃, 0.17 g) was added into 20 mL oleamine (70%) containing 0.1 mL oleic acid. The solution was heated to 180 °C under the protection of N₂, keeping this temperature for 2 h. Then the reaction was stopped and natural cooling. The precipitate was separated by centrifugal and washed with acetone five times. The obtained sample was dispersed in 30 mL hexane. To achieve an Ag/C catalyst, commercial carbon black (0.25 g) was mixed with the Ag nanoparticle dispersions by sonication. Finally, Ag/C can be collected after

centrifugation and vacuum drying. The obtained Ag/C undergoes the same electrochemical activation step as AC-Ag preparation (cyclic voltammetry scanning from 0.05 to 1.05 V (vs RHE) with five cycles, at 10 mV s$^{-1}$).

Cu/Cu$_2$O catalyst: 0.1 M copper chloride (20 mL) was added dropwise to sodium hydroxide solution (0.2 M, 30 mL) under the continuous N$_2$ bubbling. Then 200 μL ethanol and 25 mL sodium borohydride solution (0.05 M) were poured quickly. After vigorous stirring for 0.5-h, Cu/Cu$_2$O was obtained by centrifugal and rinsed with DI water, then following a vacuum drying step at 80 °C.

## Physical characterization

The phase detections of the samples were performed on a D/max 2550 VB X-ray diffractometer using Cu Kα radiation (λ = 0.154 nm) and the scan speed was 10° min$^{-1}$. X-ray photoelectron spectra (XPS) were measured using an ECSALAB250Xi spectrometer with an Al Kα X-ray (1486.6 eV) radiation for excitation, and the binding energy was corrected by C 1s value of 284.6 eV. Transmission electron microscopy (TEM) with a spherical aberration corrector (HRTEM, Titan G2 60-300) was used for morphology measurements. The surface morphology of Ag foil and AC-Ag was observed by scanning electron microscopy (JEOL-6701F) at an acceleration voltage of 5 kV. The ion component in the electrolyte was tested by ion chromatography (Thermo Scientific Aquion). In situ electrochemical FTIR spectroscopic was performed using a Fourier transform infrared spectrometer (Nicolet iS50), coupling with an in-situ electrochemical reaction cell (SPEC-I, Yuanfang Co. Ltd., Shanghai, China). The catalyst ink, obtained by dispersing Ag/C in ethanol solution (950 μL) containing 50 μL Nafion binder (5 wt% aqueous solution), was dripped onto a hemicylindrical silicon prism covered with a layer of gold membrane. A platinum wire and Ag/AgCl electrode were used as counter and reference electrodes. The electrolyte was 0.5 M KCl with or without Br-EO. During the test, the electrolyte was constantly purged with CO$_2$. The background spectrum (reflectance R$_0$) was recorded at open circuit voltage. All spectra were reported as the relative change in reflectivity, ΔR/R$_0$ = (R-R$_0$)/R$_0$. The R and R$_0$ are single-beam spectra collected at the applied bias and the reference potential. Operando Raman measurements were performed using a Raman microscope (DXR2, thermo scientific) with a 532 nm laser. The measurements were carried out in a homemade three-electrode electrochemical cell. Ag/AgCl (saturated KCl) and Pt wire were used as the reference and counter electrodes, respectively. The working electrode was a glassy carbon electrode, which was connected to an electrochemical workstation (760E) and laid flat on the microscope stage. In the process of each experiment, the electrolyte was continuously circulated by a peristaltic pump.

## Electrochemical measurement

All electrochemical tests were performed by a Biologic SP-300 workstation. A typical three-electrode system in a custom-made gastight H-type two-compartment cell (30 mL, Aida Technology Development Co., Ltd, Tianjin, China) was accepted for 2-bromoethanol (Br-EO) reduction. Pristine Ag foil, AC-Ag, or carbon paper deposited Ag/C catalyst was used as a working electrode, and their geometric areas are all 1.0 cm$^2$. The mass loading of AC-Ag/C is 1 mg cm$^{-2}$. The reference and counter electrodes were Ag/AgCl (leak-free, Aida Technology Development Co., Ltd, Tianjin, China) and Pt mesh electrodes (1 × 1 cm$^2$). The electrolyte was 0.5 M KCl (pH = 5.49 ± 0.07). Br-EO was added to the catholyte with concentrations of 50 mM, 25 M, 10 mM, 5 mM, and 2 mM. A proton exchange membrane (Nafion 117, 2.0 × 2.0 cm, 118.0 μm) was placed between the cathode and anode chamber as a separator. The catholyte was purged with CO$_2$ or N$_2$ gas (99.999%) for at least 30 min, and the gas was constantly bubbled through the catholyte during electrolysis to flow the reductive product into gas chromatography (GC), at a flow rate of 30 sccm which was controlled by a mass flow controller (HORIBA METRON). Linear sweep

voltammetry and potentiostatic electrolysis were conducted with the IR compensation of 85% in situ. To avoid the electrooxidation of metallic Ag, the onset potential for the linear sweep voltammetry was set as −0.25 V, which is far lower than the oxidation potential of Ag. The measurement in 0.5 KBr (pH = 5.47 ± 0.1) was performed once. The stability of the catalyst was evaluated by injecting Br-EO into the electrolyte after the ethylene selectivity dropped. As for the concentration dependence of Br-EO, KCl, and H$_2$O on the activity of ethylene formation, potentiostatic electrolysis was performed at −0.28, −0.38, and −0.48 V. To quench the possible generated *H, 50 mM n-butyl alcohol was added to 0.5 M KCl containing 50 mM Br-EO and the selectivity and current density was monitored at −0.38 and −0.48 V. Herein, all potentials were converted to RHE reference scale using the following equation:

$$E(\text{vs RHE}) = E(\text{vs Ag/AgCl}) + 0.197V + 0.0591 \times pH \quad (1)$$

The electrochemical active surface area (ECSA) was analyzed by Pd underpotential deposition. The electrolyte consists of a KNO$_3$ (0.1 M)/HNO$_3$ (0.01 M) aqueous solution containing 5 mM Pb(ClO$_4$)$_2$. The theoretical deposition potential for Pb$^{2+}$ is calculated to be −0.14 V (vs RHE) based on the Nernst equation. A typical three-electrode system was employed with the reference electrode and counter electrode of Ag/AgCl and Pt mesh (1 × 1 cm$^2$), respectively.

The activation energy of Br-EO reduction was acquired by linear fitting of the natural logarithm of ethylene partial current density versus the inverse temperatures based on the Arrhenius equation.

$$\ln(i) = \frac{-E\alpha}{RT} + \ln(A) \quad (2)$$

Where i is the partial current density, Eα is the activation energy, R is the gas constant, T is the reaction temperature, and A is a pre-exponential factor.

Electrochemical Impedance spectroscopy (EIS) was collected by a three-electrode cell. The measurements were conducted at constant potentials in the frequency range from 50 MHz to 100 KHz with an AC amplitude of 10 mV. Then the relaxation time constant (τ$_0$) was acquired by complex capacitance (C') analysis.

$$C'(\omega) = \frac{-Z''(\omega)}{\omega|Z(\omega)|^2} \quad (3)$$

$$C''(\omega) = \frac{Z'(\omega)}{\omega|Z(\omega)|^2} \quad (4)$$

$$Z(\omega) = \frac{1}{j\omega C(\omega)} \quad (5)$$

Where Z(ω) is impedance and ω is the penetration depth. C'(ω) and C''(ω) are the real part and imaginary part of the capacitance. According to the peak frequency (f$_0$), the τ$_0$ can be calculated using the equation of τ$_0$ = (2πf$_0$)$^{-1}$.

The transient kinetics measurement was collected by pulse amperometry. The adsorption of Br-EO was conducted at −0.08 V (vs RHE) for 100 s, followed by the application of a pulse potential of −0.28 V (vs RHE) for 10 s. The time duration of 0–2 s was analyzed for the transient kinetics.

The electrical work was calculated via the following equation.

$$W_{Br-EO} = U \times \frac{10^6}{M_{ethylene}} \times \frac{F \times N}{FE_{Br-EO}} \quad (6)$$

$$W_{ethylene} = U \times \frac{10^6 \times F \times N}{M_{ethylene} \times FE_{ethylene}} \qquad (7)$$

Where U is cell potential which was collected by two-electrode measurement using constant current mode, $M_{ethylene}$ is the molecular weight of ethylene. F is the Faradaic constant. N is the number of electon transferred.

The $CO_2$ electroreduction activity was measured by a cation-exchange membrane-separated flow cell. The catholyte is 1 M KOH (pH = 13.35 ± 0.09) and the anolyte is 1 M KBr (pH = 5.07 ± 0.03). Cu/$Cu_2O$ was first dispersed in ethanol and ultrasonic dispersion to give an ink, which was further spray coated onto a gas diffusion electrode (Sigracet 35 BC) with a geometric surface area of $2 \times 3\, cm^2$. The mass loading of the catalyst was $1\, mg\, cm^{-2}$. The Pt plate ($2 \times 3\, cm^2$) and Ag/AgCl electrode acted as counter electrode and reference electrode, respectively. During the electrochemical testing, $CO_2$ flowed over the cathode GDL with a flow rate of 50 sccm, and the electrolyte was circulated at a flow rate of $1\, mL\, min^{-1}$. The outlet gas of the cathode flowed through the anolyte jar to convert the ethylene in the gas mixture to Br-EO. After $CO_2RR$ testing, the alkaline catholyte was gradually added into the anolyte to remove the residue $Br_2$ and neutralize the formed acid, until a neutral solution formed, and then the solution was used for further reduction by the Ag/C catalyst in an H-type cell. The combined measurement for indirect $CO_2$-to-ethylene conversion was performed once as the proof of concept.

## Product analysis

The reduction products were analyzed by an on-line gas chromatograph (GC, Shimadzu, Model 2014) and $^1H$ NMR (Bruker, 400 M Hz) spectroscopy. For gas detection, the products were brought into the GC by $CO_2$ or $N_2$ with the flow rate of 30 sccm which was controlled by a mass flowmeter. The GC is equipped with one TCD detector for hydrogen and CO, one flame ionization detector (FID) coupled with a methanizer for CO and $CH_4$ detection, and one FID for $C_{2+}$ chemicals. The carrier gas was Ar (99.999%). The products collected at 1000 s were sampled into the gas sampling loop of GC (1 mL). The Faradaic efficiency and partial current density were calculated as below:

$$FE_x = \frac{nFP_0}{RT} \times \frac{1}{\alpha_x} \times peak\ area \times flow\ rate \times \frac{1}{I} \qquad (8)$$

$$J_x = FE_x \times j_{total} \qquad (9)$$

Where n is the electron transfer number. F is faraday constant ($96485\, C\, mol^{-1}$) and R is gas constant ($8.314\, J\, mol^{-1}\, K^{-1}$). $P_0$ is 1.013 bar and $T_0$ is 273.15 K. $\alpha$ is conversion factors obtained based on calibration of the GC with a standard sample.

The purity of ethylene specified on the mass basis was calculated by the following equation.

$$P_{ethylene}(wt\%) = \frac{FE_{ethylene} \times M_{ethylene}}{FE_{ethylene} \times M_{ethylene} + FE_{H2} \times M_{H2}} \qquad (10)$$

Where M is the molar molecular weight. The obtained $P_{ethylene}$ was corrected by the vapor in the product stream (Supplementary Note 11).

For $^1H$ NMR measurements, standards with known concentrations of products (0.25, 0.5, 1, 1.5, and 2.0 mM) were prepared in 1 mL 0.5 M KCl solution which contains 0.2 mL $D_2O$ and a known concentration of dimethyl sulfoxide (DMSO, DMSO: $H_2O$ = 1:1000 (V: V)) which acting as an internal reference. For the sample preparation, typically, after a bulk electrolysis experiment, 0.5 mL electrolyte, 0.1 mL $D_2O$, and 0.1 mL DMSO/$H_2O$ were mixed in an NMR tube. The NMR was performed with the water suppression method.

$$FE_y = \frac{Q_y}{Q_{total}} \times 100\% = \frac{n_y \times N \times F \times 100\%}{Q_{total}} \qquad (11)$$

n is the amount of product. $Q_{total}$ is the total transfered charge.

## DFT calculations

Density functional theory (DFT) calculations were performed with the spin-polarized generalized gradient approximation (GGA) of Perdew, Burke, and Ernzerhof (PBE) as implemented in the Vienna Ab initio Simulation Package (VASP)[56,57]. The van der Waals (vdW) energy correction was carried out with the DFT-D3 method[58]. Core electrons were handled using the projector augmented wave (PAW) method with an energy cutoff of 450 eV[59]. The electronic energy was considered self-consistent when the energy change was smaller than $10^{-5}$ eV. The geometries were relaxed until the energy difference was smaller $0.02\, eV\, Å^{-1}$. During the relaxation, the Brillouin zone with a $2 \times 2 \times 1$ Gamma-centered grid was used. A vacuum layer of 15 Å along the c-axis was used to eliminate the artificial interactions between periodic images. Spin-polarized calculations were performed for this calculation. The models based on the (111), (200), and the corresponding Cl-incorporated plane were built for free energy calculations.

The reaction mechanisms for ethylene generation are used below.

$$HO - CH_2CH_2 - Br + * \rightarrow *OHCH_2CH_2Br$$

$$*HOCH_2CH_2Br + e^- \rightarrow *OHCH_2CH_2 + Br^-$$

$$*HOCH_2CH_2 + e^- \rightarrow *CH_2CH_2* + OH^-$$

$$*CH_2CH_2* \rightarrow CH_2CH_2 + *$$

The asterisk denotes the site of the catalyst and surface adsorption.

## Data availability

The data that supports the findings of the study are included in the main text and supplementary information files. Source Data files are available in Figshare under the accession code (https://doi.org/10.6084/m9.figshare.26061289). Source data are provided in this paper.

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

## Acknowledgements

This work was financially supported by the Intergovernmental International Science and Technology Innovation Cooperation Program of National Key Research and Development Program (Grant No. 2022YFE0120200 (S.G.Z.)), the National Natural Science Foundation of China (Grant No. 21872046 (Y.Z.), 52072118 (S.G.Z.), and 52102041 (W.P.N.)), the Jiebang Guashuai Project of Hunan Province (Grant No. 2021GK1230 (S.G.Z.)), and the Natural Science Foundation of Hunan Province (Grant No. 2020JJ4174 (Y.Z.), 2022JJ40073 (W.P.N.)).

## Author contributions

S.G.Z. directed the project. W.P.N. conceived the idea and designed the experiments. W.P.N. and W.Z. collected the electrochemical results. H.J.C. and N.Z.T. performed the in situ testing. W.P.N., T.H., and Y.Z. analyzed the experimental data. W.P.N. wrote the paper and S.G.Z. revised the paper.

## Competing interests

The authors declare no competing interests.
