## [Peer Review File · Nature Communications]

REVIEWER COMMENTS

Reviewer #1 (Remarks to the Author):

The manuscript by W. Ni et al. introduces an indirect electrochemical CO₂ reduction approach for the generation of high-purity ethylene. From my point of view, the practical application prospects of this method are subject to debate, such as the low current density, and the claimed advantages in energy consumption. The scenario of lower energy consumption is created by the relatively lower cell voltage at the low current density, without considering the production efficiency for practical purposes (as mentioned in Q2 by Reviewer #1 previously). Nevertheless, this study does offer a new strategy for high-purity ethylene preparation. With the current convincing results and discussions, it is considered suitable for publication.

Reviewer #4 (Remarks to the Author):

The manuscript entitled “High-purity Ethylene Production via Indirect Carbon Dioxide Electrochemical ” from Zhang et al reports an indirect reduction strategy for CO₂-to-ethylene conversion, one that employs 2-bromoethanol (Br-EO) as a mediator. Br-EO is initially generated from CO₂RR and subsequently undergoes reduction to ethylene without the need for energy-intensive separation steps. They claimed that through this indirect strategy, ethylene with the highest purity of 99.9wt% was obtained at -0.48 V from the electrolyte after CO₂ reduction over the Cu/Cu₂O catalyst and this contribution brings new insight into the design of CO₂ electroreduction technologies. However, some important details are missing in the paper, such as whether acidic conditions have a greater impact on the production of hydrogen and whether liquid products are ignored be worthy of discussion. Moreover, in my opinion, the ethylene spilled from the electrolyte could definitely not eliminate water vapor to reach such high purity as they claimed. Thus, the poorly organized manuscript make it fail to meet the standard for a publication in Nature Communication.

My comments are as follows:

1. Lines 114-115 suggest that the main product is ethylene, with a small amount of hydrogen derived from water. In my opinion, the anodic electrolyte should be acidic after the reaction of the first CO₂RR cell. As the author claimed, such an anodic electrolyte was directly used for the following reduction reaction. It is well-known that the hydrogen evolution reaction (HER) would be boosted under acidic media, the author should pay more attention to the abnormal HER suppression phenomenon and explain it.

2. Using KCl solution to simulate the electrolyte after CO₂RR is ridiculous due to that the anodic electrolyte much contain Br⁻ ion to produce Br-EO from the first CO₂RR cell. Besides, the author carried out performance evaluation under weak base and strong base conditions, which is unnecessary and meaningless. It should be addressed that the anodic electrolyte would become acidic after long-term electrolysis even if using a buffer solution. So, why did the author do some needless control experiments without carrying out the significant comparison? The conclusion from such a confused experiment could not convince the reviewer.

3. In addition, it is reported that the halogen substituted ethanol could be easily converted to ethylene oxide (EO) under alkaline conditions (Science 2020, 368,1228–1233). However, the author failed to carry out such discussion. The reviewer wonders that if the EO is observed in the experiment carried out in the simulated alkaline environment or not?

4. In this work, the author claims that the indirect pathway can avoid water separation compared with the direct pathway and that the ethylene produced by the indirect pathway is almost pure. However, the reduction of Br-EO is performed in a single cell, it means that the generated C₂H₄ must across through the aqueous electrolyte to be collected, and how does the water avoid being carried out?

Reviewer #1:

The manuscript by W. Ni et al. introduces an indirect electrochemical CO₂ reduction approach for the generation of high-purity ethylene. From my point of view, the practical application prospects of this method are subject to debate, such as the low current density, and the claimed advantages in energy consumption. The scenario of lower energy consumption is created by the relatively lower cell voltage at the low current density, without considering the production efficiency for practical purposes (as mentioned in Q2 by Reviewer #1 previously). Nevertheless, this study does offer a new strategy for high-purity ethylene preparation. With the current convincing results and discussions, it is considered suitable for publication.

Answer: Thank you for agreeing to accept our manuscript. We concur with the observation that the performance presented in this work falls short of the requirements for practical application. The principal innovation lies in offering an alternative strategy to conventional direct CO₂ electroreduction for producing high-purity ethylene. To achieve the production efficiency necessary for practical purposes, further research on catalyst design to enhance the selectivity of CO₂-to-Br-EO conversion is required. Additionally, improvements in the flow cell design are essential to increase the current density of Br-EO reduction.

Reviewer #4:

The manuscript entitled “High-purity Ethylene Production via Indirect Carbon Dioxide Electrochemical Reduction” from Zhang et al reports an indirect reduction strategy for CO₂-to-ethylene conversion, one that employs 2-bromoethanol (Br-EO) as a mediator. Br-EO is initially generated from CO₂RR and subsequently undergoes reduction to ethylene without the need for energy-intensive separation steps. They claimed that through this indirect strategy, ethylene with the highest purity of 99.9wt% was obtained at -0.48 V from the electrolyte after CO₂ reduction over the Cu/Cu₂O catalyst and this contribution brings new insight into the design of CO₂ electroreduction technologies. However, some important details are missing in the paper, such as whether acidic conditions have a greater impact on the production of hydrogen and whether liquid products are ignored and are worthy of discussion. Moreover, in my opinion, the ethylene spilled from the electrolyte could not eliminate water vapor to reach such high purity as they claimed. Thus, the poorly organized manuscript makes it fail to meet the standard for a publication in Nature Communication.

1. Lines 114-115 suggest that the main product is ethylene, with a small amount of hydrogen derived from water. In my opinion, the anodic electrolyte should be acidic after the reaction of the first CO₂RR cell. As the author claimed, such an anodic electrolyte was directly used for the following reduction reaction. It is well-known that the hydrogen evolution reaction (HER) would be boosted under acidic media, the author should pay more attention to the abnormal HER suppression phenomenon and explain it.

Answer: Thank you for your insightful comment. We concur with the reviewer’s comment that the anodic electrolyte may become acidic after the reaction in the first CO₂RR cell, which would facilitate the hydrogen evolution reaction. However, our investigation revealed that the anolyte is

only mildly acidic after electrolysis, with a pH of 5.80. Consequently, the reduction of Br-EO to ethylene remains the predominant reaction in this weakly acidic electrolyte, with FE_{ethylene} higher than 98.0%. Therefore, significant hydrogen evolution was not observed. Additionally, it is important to clarify that we did not directly use the anodic electrolyte post-reaction for the conversion of Br-EO to ethylene. Instead, the anolyte was neutralized by the alkaline catholyte before being introduced into the second electrolytic cell for ethylene generation, resulting in a neutral environment within our system. In the Experimental Section of our original manuscript, we described this process as follows: “After CO₂RR testing, **the anolyte was mixed with the catholyte**, and then the electrolyte was directly used for further reduction by the Ag/C catalyst in an H-type cell”.

The detailed responses are presented as outlined below.

- **The anolyte is a weak acid after electrolysis, partially due to the proton transport over the cation exchange membrane.** As shown in **Figure R1a**, we first monitored the pH fluctuations of the anolyte post-electrolysis. A solution with saffron yellow was observed in the anolyte after 2 hours of electrolysis at -200 mA cm^{-2} , registering a pH of 5.80, indicative of a mild acidic environment. After standing for 19 hours, the pH of the anolyte increased to 6.10, subsequently ascending to 6.25 and 6.54 at 38 and 57 hours, respectively. The observed mild acidity, rather than a robust acidic environment, was partially ascribed to the cation exchange membrane (CEM) used in the electrolytic cell. The protons, that are generated in the anolyte, can partially traverse the CEM and interact with hydroxide ions (OH^-) present in the catholyte. To substantiate this assertion, we established an electrolysis system featuring a cathodic reaction of CO₂ reduction and anodic oxidation of ferrocyanide ions ($\text{Fe}(\text{CN})_6^{4-}$) (**Figure R1b**). Notably, the oxidation of $\text{Fe}(\text{CN})_6^{4-}$ does not entail the generation or depletion of protons, hence its corresponding influence on the pH variations of the anolyte can be discounted. The aqueous solution of 0.5 M KCl containing 25 mM H₂SO₄ was used as the anolyte (pH = 1.52, as presented in the upside section of **Figure R1c**). On the cathodic side, CO₂ can react with KOH in the catholyte and generate KHCO₃. The resultant HCO₃⁻ may permeate the CEM and further interact with the protons present within the anolyte, potentially modulating its pH. Consequently, we scrutinized the pH of the anolyte following a 10-hour incubation period with continuous CO₂ flow through the cathode chamber. The pH of the anolyte was detected to be 1.54 (downside section of **Figure R1c**), almost mirroring that of the pristine electrolyte. This outcome attests to the marginal impact of HCO₃⁻ crossover on the pH dynamics of the anolyte within our experimental framework. Conversely, after the electrolysis of the CO₂ - $\text{Fe}(\text{CN})_6^{4-}$ system under -50 mA cm^{-2} for 2 hours, the anolyte was collected, revealing a corresponding pH of 2.22 as indicated by the pH meter (**Figure R1d**). This means the electric field-driven proton crossover can significantly buffer the pH reduction of anolyte, thereby preventing the formation of a strong acid environment.
- **Br-EO-to-ethylene conversion is still the primary reaction in weak acidic electrolytes ($FE_{\text{ethylene}} > 98.0\%$), attributing to the cation effect and the superior intrinsic activity for Br-EO reduction over Ag.** As delineated in **Figure R1e**, we endeavored to elucidate the impact of an acidic environment on the selectivity of Br-EO electroreduction. The pH of a 0.5 M KCl aqueous solution was modulated by adding H₂SO₄ across varying concentrations. In a dilute acidic KCl aqueous solution ($C_{\text{H}_2\text{SO}_4} = 0.001 \text{ mM}$, pH = 5.7), the FE_{ethylene} still reached

98.6%. With the proton concentration increased to 50 mM (equal to the concentration of added Br-EO, pH = 1.59), we also observed a notable ethylene selectivity of 88.5%, and the FE_{ethylene} remaining above 50.0% even upon introducing 0.1 M H_2SO_4 . In the electrolyte comprising 0.5 M KCl + 25 mM H_2SO_4 + 50 mM Br-EO, ethylene emerges as the principal product across the potential window ranging from -0.228 to -0.528 (**Figure R1f**). These findings underscore the feasibility of achieving high-purity ethylene production within a mild acidic solution.

To disclose the origin of the high ethylene selectivity in a weak acid solution, we initially assessed the electrochemical performance of Br-EO reduction in a pure acidic solution (25 mM H_2SO_4) devoid of any added metal salt, intending to investigate the potential cation effect reported in CO_2 electroreduction in acidic electrolytes (*Science*, 2021, **372**, 1074; *Nat. Catal.*, 2022, **5**, 268). The LSV curves evince a positive shift in the onset potential and an augmented total current density (**Figure R1g**). A greater H_2 selectivity was discerned, exemplified by the FE_{H_2} reaching 23.8% at -0.328 V, surpassing that achieved in 0.5 M KCl + 25 mM H_2SO_4 + 50 mM Br-EO (11.5%). This means the K^+ benefits the suppression of hydrogen evolution reaction (HER), plausibly attributable to the altered interfacial electric field hindering proton migration. Intriguingly, even under the pure acid electrolyte, Br-EO reduction remains the primary reaction within the range of -0.238 to -0.428 V (**Figure R1h**). Therefore, it can be speculated that the AC-Ag electrode exhibits superior intrinsic activity for Br-EO reduction. According to the dependence between overpotential and $\log j$, the exchange current density (j^0), a key parameter for evaluating the catalytic ability of an electrode toward different reactions, can be calculated for HER and Br-EO reduction. As shown in **Figure R1i**, the linear relationships were observed, with the equations $y = 0.221x + 0.259$ and $y = 0.149x + 0.504$ for H_2 and ethylene formation, respectively. The corresponding j^0 was calculated to be $6.8 \times 10^{-2} \text{ mA cm}^{-2}$ and $4.16 \times 10^{-4} \text{ mA cm}^{-2}$ for Br-EO reduction and HER, respectively. These outcomes further bolster the assertion that the AC-Ag electrode is predisposed to promote Br-EO reduction, elucidating the observed primary ethylene formation in mild acidic electrolytes.

- **The mildly acidic anolyte was not directly utilized for the subsequent reduction reaction; instead, it was neutralized by the alkaline catholyte during our experimental procedure.** In the Experimental Section of our original manuscript, the stated procedure reads as follows: “After CO_2RR testing, **the anolyte was mixed with the catholyte**, and then the electrolyte was directly used for further reduction by the Ag/C catalyst in an H-type cell”. In detail, **the alkaline catholyte is gradually added into the anolyte, to remove the residue Br_2 and neutralize the formed acid**, transforming the solution from saffron yellow to colorless, as depicted in the inset of **Figure R1j**. Notably, the gradually added alkaline catholyte did not induce any changes in the Br-EO in the anolyte, as confirmed by the 1H NMR (**Figure R1j**). Subsequently, the electrolysis of this mixed solution by AC-Ag electrode gave the exclusive ethylene peak in the online GC, with the FE_{ethylene} exceeding 95.0%, with no identifiable peak for H_2 (retention time of 2.25 min) observed (**Figure R1k-R1l**). These results further support that neutralizing anolyte by the catholyte of the first electrolytic cell can effectively remove the potential influence of a weak acid environment.

In summary, the generated Br_2 induces the gradual acidification of the anolyte within the first

electrolytic cell, giving a weakly acidic environment. However, the inherent superior activity of the AC-Ag electrode for Br-EO reduction to HER engenders the high selectivity of ethylene in this weak acidic solution. Moreover, the weak acidic anolyte was neutralized by the alkaline catholyte in our work, giving a neutral solution for Br-EO electrochemical reduction. Thus, we did not observe the pH-relevant variation in hydrogen evolution activity. **To address the reviewer's concern, we have made the following modifications.** (1) The corresponding expression in the Experimental Section was corrected, "After CO₂RR testing, the alkaline catholyte was gradually added into the anolyte to remove the residue Br₂ and neutralize the formed acid, until a neutral solution formed." (2) We note that the schematic diagram (**Figure 1b**) did not incorporate the neutralization step of the anolyte, potentially leading to misconceptions regarding our indirect reduction pathway. Consequently, **Figure 1** has been revised accordingly. (3) The corresponding figures and discussions about the potential pH effect of anolyte have been added to the revised Supporting Information.

Figure R1. (a) The color and pH of anolyte in the first electrolytic cell after standing for different intervals. (b) Electrolytic system design for evaluating the influence of cation exchange membrane on the pH variation of anolyte. The indication of pH meter for (c) 0.5 M KCl + 25 mM H₂SO₄(upside), 0.5 M KCl + 25 mM H₂SO₄ with CO₂ flow through the cathode chamber for 10 h (downside), and (d) 0.5 M KCl + 25 mM H₂SO₄ + 1 M KFe(CN)₆ after electrolysis. (e) FE of H₂ and ethylene for AC-Ag electrode collected in 0.5 M KCl + 50 mM Br-EO with different concentrations of H₂SO₄ at -0.328 V. (f) FE of H₂ and ethylene for AC-Ag electrode collected in 0.5 M KCl + 25 mM H₂SO₄ + 50 mM Br-EO from -0.228 to -0.528 V. (g) LSV curves collected in 25 mM H₂SO₄ with and without 50 mM Br-EO. (h) FE of H₂ and ethylene for AC-Ag electrode in 25 mM H₂SO₄ containing 50 mM Br-EO from -0.228 to -0.528 V. (i) The dependence of overpotential for HER and Br-EO reduction versus log j. (j) The ¹H NMR of Br-EO in the pristine anolyte and after neutralization with KOH catholyte. Inset is the photo of the anolyte after neutralization. Online GC spectra of (k) FID, (l) TCD detector for Br-EO electroreduction in the neutralized anolyte.

2. Using KCl solution to simulate the electrolyte after CO₂RR is ridiculous due to that the anodic electrolyte much contain Br⁻ ion to produce Br-EO from the first CO₂RR cell. Besides, the author carried out performance evaluation under weak base and strong base conditions, which is unnecessary and meaningless. It should be addressed that the anodic electrolyte would become acidic after long-term electrolysis even if using a buffer solution. So, why did the author do some needless control experiments without carrying out the significant comparison? The conclusion from such a confused experiment could not convince the reviewer.

Answer: Thank you for your comment. The structure of this paper is delineated as follows (**Figure R2a**). First, an indirect method for producing high-purity ethylene from CO₂ is introduced, utilizing Br-EO as a crucial intermediary. Specifically, CO₂ is initially transformed into Br-EO, which is then reduced to ethylene. Given the established conversion of CO₂ to Br-EO, we initially assessed the feasibility of converting Br-EO to ethylene using an Ag catalyst. To enhance the electrochemical activity of the Br-EO reduction, modifications were made to Ag-based catalysts, and the corresponding mechanisms were investigated. Ultimately, after optimizing the catalyst, we evaluated the performance of the integrated CO₂-to-Br-EO and Br-EO-to-ethylene system for high-purity ethylene production. Consequently, our work is divided into two parts. **Part I** focuses on catalyst design and mechanistic analysis for the Br-EO-to-ethylene conversion (highlighted in red in **Figure R2a**). **Part II** details the development of the integrated system for high-purity ethylene production (highlighted in blue in **Figure R2a**).

In **Part I**, all experiments were conducted in a single electrolytic cell (**Figure R2b**). Commercial Br-EO was used as the reactant, and Ag-based catalysts (Ag foil, AC-Ag, and AC-Ag/C) served as the cathode. To prevent interference in detecting Br-containing products from Br-EO reduction, a KCl aqueous solution was employed as the electrolyte. Additionally, the electrochemical performance of Br-EO reduction was assessed in KHCO₃ and KOH to evaluate the general efficacy of Ag-based catalysts in converting Br-EO to ethylene.

In **Part II**, an integrated system comprising two electrolytic cells was constructed (**Figure R2c**). The first cell, an H-type flow cell, was used for the reduction of CO₂ to produce Br-EO in the

anolyte. The second cell, equipped with the AC-Ag/C catalyst, was used for the conversion of Br-EO to ethylene. In this cell, the neutralized anolyte from the first cell served as the electrolyte, and Br-EO, derived from CO₂ reduction, was used as the reactant.

Based on the aforementioned descriptions, the use of KCl, KHCO₃, and KOH electrolytes is specific to **Part I** of this work. In contrast, **Part II** involves the formation of Br-EO from CO₂ and the acidification of the anolyte in the first electrolytic cell. Therefore, the electrolytes used in **Part I** do not impact the formation of Br-EO from CO₂RR in **Part II**, which employs a KBr anolyte. Similarly, the use of KHCO₃ and KOH electrolytes in **Part I** does not affect the pH of the anolyte in the first electrolytic cell of **Part II**.

The detailed responses are presented as follows.

Figure R2. (a) Descriptions for the structure of this work. (b) Schematic illustration for the electrolytic cell used for explorations of Part I of this work (catalyst design and mechanism analysis for Br-EO-to-ethylene conversion). (c) Schematic illustration for the electrolytic cell used for explorations of Part II of this work (integrated system for high-purity ethylene production).

- **Regarding the use of KCl solution, our response is as follows:** In the integrated CO₂-to-Br-EO and Br-EO-to-ethylene system (**Part II** of this work), we did not use KCl solution in either the first electrolytic cell (CO₂-to-Br-EO system with KBr solution as the anolyte and KOH solution as the catholyte) or the second electrolytic cell (Br-EO-to-ethylene system,

where the electrolyte is the anolyte from the CO₂-to-Br-EO reaction neutralized by the catholyte, as explained in our response to the first question). We only employed KCl when studying the Br-EO-to-ethylene reaction mechanism independently (**Part I** of this work). This was because Br ions from Br-containing electrolytes (such as KBr solution) can complicate the identification of the Br-EO reduction product due to the involvement of C-Br bond cleavage in the Br-EO-to-ethylene conversion. In fact, since the electrolyte for the Br-EO-to-ethylene reaction originates from the anodic electrolyte of CO₂-to-Br-EO, we have considered various electrolytes such as alkaline (e.g., KOH), neutral (e.g., KHCO₃, KCl), and acidic (H₂SO₄) solutions as candidates for conducting the mechanistic analysis of the Br-EO-to-ethylene reaction. However, only neutral electrolytes align with the research objectives of this work. As shown in **Figure R3a**, as for alkaline electrolytes, CO₂ would react with OH⁻ in the bulk of electrolytes, given the utilization of an H-type cell. Concerning acidic electrolytes, there are two potential proton sources, H⁺ and H₂O molecules. It is better to utilize electrolytes with exclusive proton sources to simplify the mechanism exploration. In this context, a neutral electrolyte, excluding KHCO₃ (HCO₃⁻ also serves as a proton donor (*J. Am. Chem. Soc.* 2017, **139**, 17109)), emerges as the optimal choice for our inquiry. Therefore, a KCl electrolyte was used for the mechanistic analysis of Br-EO reduction. However, in the integrated CO₂-to-Br-EO and Br-EO-to-ethylene system, KBr anolyte was used for producing Br-EO in the first electrolytic cell. Please note that replacing the electrolyte from KBr to KCl does not affect the high ethylene selectivity. As depicted in **Figure R3b**, the FE_{ethylene} exceeds 97.0% for the AC-Ag/C electrode over the potential range from -0.08 V to -0.58 V in the 0.5 M KBr aqueous solution.

In response to the reviewer's concerns, we have added **Figure R3a** to the revised supporting information, illustrating the selection of KCl electrolytes. Additionally, **Figure R3b**, which depicts the electrochemical performance of AC-Ag/C in 0.5 M KBr, has been included as **Figure 3f** in the revised manuscript.

- **Regarding the use of weak base (KHCO₃ solution) and strong base (KOH solution) conditions, our response is as follows:** We only employed KHCO₃ and KOH solutions for investigating the Br-EO-to-ethylene reaction mechanism independently (**Part I** of this work), prior to studying the integrated CO₂-to-Br-EO and Br-EO-to-ethylene system. However, when coupled with the CO₂-to-Br-EO reaction (**Part II** of this work), the Br-EO-to-ethylene reaction is conducted in a neutral electrolyte (as mentioned in our response to the first question). As presented in lines 223-231, the rationale for using weak base and strong base conditions in the independent study of the Br-EO-to-ethylene reaction mechanism (Part I of this work) can be succinctly encapsulated as follows: (1) To investigate the universal nature of the high activity observed in the conversion of Br-EO to ethylene over the Ag electrode, the consistent high ethylene selectivity across different electrolytes substantiates the widespread ability to produce high-purity ethylene. (2) To explore the potential pH effect on Br-EO reduction. It is known that a proton-involved rate-determining step can induce pH-dependent activity variations. The nearly identical current densities upon switching from 0.1 M KHCO₃ to 0.1 M KOH electrolytes (at the potentials with respect to SHE, **Figure R3c**) corroborate the proton-independent nature of Br-EO reduction. This finding aligns with the reaction kinetics analysis (**Figure 4b** and **4c** in the manuscript).

Considering the misunderstanding mentioned by the reviewer and the competitive reaction between Br-EO and OH⁻ in the KOH solution (details can be found in the answer to the Question 3), we deleted the corresponding results and discussions regarding Br-EO reduction in both 0.1 M KHCO₃ and 0.1 M KOH in the revised manuscript.

It is essential to underscore that the proposed revisions will uphold the conclusions drawn in our original manuscript. This is due to two main reasons: Firstly, the primary findings, which encompass the enhanced intrinsic activity of electrochemically activated Ag electrodes, the first-order dependency of Br-EO concentration and electrochemical activity, the adsorption/desorption dynamics of reactants and intermediates, and the comprehensive delineation of reaction mechanisms, are derived from analyses conducted in KCl electrolytes rather than in 0.1 M KHCO₃ and 0.1 M KOH solutions. Secondly, the proton-independent nature of Br-EO reduction was deduced from electrochemical tests carried out in 0.1 M KHCO₃ and 0.1 M KOH solutions. Moreover, such a proposition can be substantiated by the minimal correlation observed between J_{ethylene} and H₂O concentration, as well as the insensitivity to a proton-quenching agent.

Figure R3. (a) The selection route for the electrolyte. (b) FE of H₂ and ethylene of AC-Ag/C electrode in 0.5 M KBr electrolyte. (c) The partial current density of ethylene in 0.1 M KHCO₃ and 0.1 M KOH plotted with respect to the SHE.

3. In addition, it is reported that the halogen substituted ethanol could be easily converted to ethylene oxide (EO) under alkaline conditions (Science 2020, 368,1228–1233). However, the author failed to carry out such discussion. The reviewer wonders that if the EO is observed in the experiment carried out in the simulated alkaline environment or not?

Answer: Thank you for this insightful comment. Firstly, we only used KOH solution as the

alkaline electrolyte in our mechanistic studies of the Br-EO-to-ethylene reaction, but we did not use alkaline electrolytes in the integrated CO₂-to-Br-EO and Br-EO-to-ethylene system, where we employed a neutral electrolyte and did not observe the formation of ethylene oxide. Secondly, in line with the reviewer's comments, we found that Br-EO can react with KOH to form ethylene oxide in alkaline electrolytes. However, this chemical reaction is time-dependent; during long-term electrolysis, most Br-EO undergoes chemical transformation to ethylene oxide, leaving less Br-EO available for electrochemical reduction to ethylene, thus reducing the electrochemical conversion efficiency of Br-EO. During short-term electrolysis, the concentration of Br-EO available for the electrochemical reaction remains high enough for obtaining a high FE_{ethylene}. Consequently, the hydrogen evolution reaction (HER) is not the predominant electrode reaction. Indeed, our mechanistic studies were conducted under short-term electrolysis conditions, not long-term electrolysis.

The detailed responses are presented as follows.

- At first, it is important to clarify that we only used KOH solutions to investigate the Br-EO-to-ethylene reaction mechanism independently before investigating the combined CO₂-to-Br-EO and Br-EO-to-ethylene system. In this integrated system, the electrolyte is derived from the anodic electrolyte produced from the CO₂-to-Br-EO reaction and neutralized by the cathodic electrolyte thereafter. After electrolysis, the organic phase of the electrolyte was extracted with toluene for GC analysis. Notably, in the neutral electrolyte of the integrated indirect CO₂-to-ethylene process, ethylene oxide was not detected by GC and NMR analysis. As shown from the GC spectra in **Figure R4a**, the peak corresponding to ethylene oxide was notably absent (depicted by the blue line), indicating its absence in the electrolyte. Additionally, the ¹³C NMR spectrum of the electrolyte post-electrolysis only showed signals attributed to Br-EO, with no characteristic peak for ethylene oxide (**Figure R4b**). These findings indicate that Br-EO does not convert to ethylene oxide in our indirect CO₂-to-ethylene system (including CO₂-to-Br-EO and Br-EO-to-ethylene processes).
- Secondly, in our investigation of the Br-EO-to-ethylene reaction mechanism independently, we reassessed it in a KOH solution, confirming the formation of ethylene oxide. The electrolyte composed of 0.1 M KOH + 50 mM Br-EO was subjected to ¹³C NMR spectroscopy analysis after electrolysis at -0.2 V (vs RHE) for 0.5 hours. As depicted in **Figure R4c**, ethylene oxide, along with residual Br-EO, was identified in the electrolyte. This validates that Br-EO can chemically convert to ethylene oxide in an alkaline environment (*Angew. Chem. Int. Ed.*, 2024, 63, e202402950; *Science*, 2020, 368, 1228), as noted by the reviewer.

For long-term electrolysis, since most Br-EO reacts chemically with KOH to form ethylene oxide, the concentration of Br-EO available for the electrochemical reaction becomes too low, thus diminishing the electrochemical conversion efficiency of Br-EO. For instance, using a FE_{H₂} greater than 50.0% as the endpoint criterion, we measured the total charge (Q_{total}) of electrolysis in 0.5 M KCl and 0.1 M KOH, each containing 25 mM Br-EO, over the AC-Ag electrode at -0.2 V (vs RHE). In these two systems, the required electrolysis time was approximately 391 minutes for 0.5 M KCl + 25 mM Br-EO (**Figure R4d**), and 172 minutes for 0.1 M KOH + 25 mM Br-EO (**Figure R4e**). The obtained Q_{total} is 132.76 C in 0.5

M KCl + 25 mM Br-EO, corresponding to a Br-EO conversion of 91.72%. However, the Q_{total} is only 97.1 C for 0.1 M KOH + 25 mM Br-EO. The difference in Q_{total} between 0.5 M KCl and 0.1 M KOH is attributed to the chemical transformation of Br-EO to ethylene oxide in the alkaline electrolyte. These results further indicate that long-term electrolysis in an alkaline electrolyte significantly reduces the electrochemical conversion efficiency of Br-EO due to its chemical reaction with KOH to form ethylene oxide.

While the chemical conversion could potentially reduce the electrochemical conversion efficiency of Br-EO during long-term electrolysis, it does not impact the achievement of high $FE_{ethylene}$ during short-term electrolysis. Firstly, the reaction between Br-EO and KOH is a non-electrochemical process that does not consume electrons, so the total detected electrons (Q_{total}) are attributed solely to the electrochemical reduction of Br-EO or the hydrogen evolution reaction. Secondly, ethylene oxide is not electrochemically reduced to ethylene on our engineered AC-Ag electrode. As shown in **Figure R4f**, the direct electrolysis of an electrolyte containing 0.1 M KOH and 50 mM Br-EO at -0.2 V initially generates a significant current due to ethylene production from Br-EO reduction (0 h in **Figure R4f**). However, after the electrolyte stands for 24 hours before electrolysis, the minimal current observed indicates that most Br-EO has reacted with KOH to form ethylene oxide, which is not further reduced. Thus, the detected ethylene is exclusively derived from the electrochemical reduction of Br-EO. According to the following equation,

$$FE_{ethylene} = \frac{nFm_{ethylene}}{M_{ethylene} \times Q_{total}}$$

neither Q_{total} nor $m_{ethylene}$ is influenced by the chemical reaction between Br-EO and KOH. Therefore, there is no correlation between $FE_{ethylene}$ and the chemical consumption of Br-EO by KOH during electrochemical reduction testing.

During short-term electrolysis, there is insufficient time for the chemical conversion of most Br-EO to ethylene oxide in KOH, thus maintaining a high concentration of Br-EO available for the electrochemical reaction and ensuring a high $FE_{ethylene}$. In our study, the electrochemical measurements for Br-EO reduction mechanism analysis were conducted using a short-term model in an alkaline environment. As shown in **Figure R4g**, ethylene selectivity was assessed at five potentials (ranging from 0.104 to -0.296 V) in 0.1 M KOH, with each potential evaluated for 10 minutes, totaling 50 minutes for all tests. This duration (50 minutes) is shorter than the time required to reduce $FE_{ethylene}$ below 96.4% (> 80 minutes, **Figure R4e**). Therefore, during the 50 minutes of electrolysis, the remaining Br-EO is sufficient to support electrochemical reduction with a high $FE_{ethylene}$.

Given the reduced conversion efficiency of Br-EO in alkaline electrolytes and the discussions presented in response to **Question 2**, we have omitted the data obtained in 0.1 M KOH from the revised manuscript. This adjustment does not affect the overall integrity of the research.

Figure R4. (a) GC spectra of toluene, toluene + Br-EO, ethylene oxide + THF, and toluene extract liquor for electrolyte after electrolysis at -0.58 V for one hour. (b) The ^{13}C NMR spectrum of 0.1 M KOH + 50 mM Br-EO after 1 h electrolysis at -0.2 V. (c) ^{13}C NMR spectra of pure Br-EO and the electrolyte after electrolysis at -0.58 V for one hour. The passed charge, time, and FE of H_2 and ethylene for AC-Ag electrode in (d) 0.5 M KCl + 25 mM Br-EO, and (e) 0.1 M KOH + 25 mM Br-EO. (f) Total current density of AC-Ag electrode collected in 0.1 M KOH + 50 mM Br-EO without standing (0 h) and after standing for 24 h (24 h), within the potential window of 0.1 - -0.3 V. (g) The current and $\text{FE}_{\text{ethylene}}$ of AC-Ag/C electrode collected in 0.1 M KOH + 50 mM Br-EO without standing.

4. In this work, the author claims that the indirect pathway can avoid water separation compared with the direct pathway and that the ethylene produced by the indirect pathway is almost pure. However, the reduction of Br-EO is performed in a single cell, it means that the generated C_2H_4 must cross through the aqueous electrolyte to be collected, and how does the water avoid being carried out?

Answer: Thank you for this comment. We appreciate the reviewer's attention to this issue. In fact, analyzing product purity with respect to water vapor influence is a common issue in all electrochemical systems involving aqueous electrolytes and warrants thorough investigation, despite being largely overlooked in previous research. Our quantitative analysis reveals that the water content at the gas outlet of the Br-EO-to-ethylene electrolytic cell is very low (0.322 wt% under the current density of -30 mA cm^{-2}). Specifically, the presence of water vapor results in only

a 0.31 wt% difference in ethylene purity. Additionally, this minimal amount of water can be effectively removed using a water-adsorption trap with color-changing silica gel, even under industrial current conditions.

The detailed responses are outlined below.

- **Qualitative analysis using color-changing silica gel demonstrated that the water content in the exhaust gas of the Br-EO electrolytic cell is minimal.** The setup for this qualitative analysis is shown in **Figure R5a**. The exhaust gas of the Br-EO-to-ethylene electrolytic cell passes through a desiccator containing 20 g of color-changing silica gel, which can adsorb at least 5 wt% water and change color from blue to pink upon water adsorption, to detect and analyze the moisture content in the gas. The feasibility of this protocol was first assessed by bubbling N₂ gas into the cathode chamber. After 115 minutes at a flow rate of 50 sccm, a recognizable pink area appeared, corresponding to an N₂ volume of approximately 5.75 L (**Figure R5b**). Reducing the flow rate to 30 and 15 sccm, with gas volumes of 6.6 and 7.65 L, extended the color-change time to 220 and 510 minutes, respectively. These results indicate that our method can qualitatively analyze vapor in the product stream at sufficiently high gas flow rates. Next, the electrolysis of Br-EO was conducted using AC-Ag as the working electrode. The silica gel color was monitored at various intervals (**Figure R5c**). Remarkably, the silica gel remained blue even after 34 hours of electrolysis, with a total charge of 3108 C (**Figure R5d**). This suggests that the water vapor in the product stream is minimal within our Br-EO-to-ethylene conversion system.
- **Quantitative analysis showed a 0.31 wt% decrease in the purity of ethylene when accounting for water vapor influence.** The setup for this analysis is depicted in **Figure R5e**, where gas from the cathode chamber initially passed through a flask containing 5.0 g of ultra-dry N,N-dimethylacetamide (DME), known for its effective water vapor adsorption. To prevent atmospheric moisture from affecting results, a tail-gas dryer was attached to the flask's opposite end. Water content in the DME was subsequently measured using a Karl-Fischer titrator to quantify water captured from the exhaust gas of the cathode chamber.

As shown in **Figure R5f**, the initial water content of the pristine DME is 254.3±31.6 ppm. After 48 hours of electrolysis at a constant current of -30 mA cm⁻², with periodic addition of Br-EO to the catholyte, the detected water content in DME increased to 1223±142 ppm (all raw data on water content testing are included in **Figure R5g**). Therefore, the net increase in water content after electrolysis is 968.7 ppm. Considering the near-complete conversion of ethylene, the water content in the electrolysis product can be calculated as follows:

$$\begin{aligned} H_2O_{wt\%} &= \frac{968.7 \times 10^{-6} \times 5.0}{\frac{0.03 \times 48 \times 3600}{96485} \times 28} \\ &= 0.322 \text{ wt\%} \end{aligned}$$

The mass fraction of H₂O in the product stream is 0.322 wt% at a current density of -30 mA cm⁻². Based on this data, we corrected the reported purity of ethylene in our manuscript using the following equation:

$$P_{ethylene}(m\%) = \frac{FE_{ethylene} \times M_{ethylene}}{\frac{FE_{ethylene} \times M_{ethylene} + FE_{H_2} \times M_{H_2}}{100 - 0.322}}$$

Using this equation, the average ethylene purity during the continuous electrolysis over 6 hours in the integrated indirect route was corrected to 98.00%, **which is 0.31 wt% lower than previously reported without accounting for water vapor in the product stream.** This difference is likely overestimated due to the current density gradually decreasing with the consumption of Br-EO, resulting in a lower water content in the gaseous product than calculated under constant current conditions.

- **The water in the product stream can be efficiently removed using silica gel.** Figure R5h illustrates the placement of a water-absorption trap filled with 30 g color-changing silica gel between the cathode chamber and the ultra-dry DME, aimed at eliminating water in the exhaust gas of the cathode chamber. After continuous electrolysis for 48 hours at -30 mA cm^{-2} , the water content was measured at $211.7 \pm 11.5 \text{ ppm}$ (electrolysis + silica gel in **Figure R5j**), comparable to pristine ultra-dry DME. Thus, the absorption trap effectively eliminates nearly all water in the product stream under our experimental conditions.

We also investigated the feasibility of water removal using color-changing silica gel under industrial current conditions. Assuming a current of 1000 A and considering nearly 100% ethylene selectivity, the ethylene production rate can be estimated as follows:

$$V_{ethylene} \left(\frac{mL}{min} \right) = \frac{1000 \times 60}{96485} \times 8.314 \times 298 \times 1000$$

$$\approx 15 \text{ mL/min} \approx 15 \text{ sccm}$$

At a constant flow rate of 15 sccm, the water content in the DME absorber without color-changing silica gel was measured at $4171.5 \pm 13.5 \text{ ppm}$ after bubbling for 2 hours (**Figure R5f**, 15 sccm + DME), indicating a gas water content of 0.72 wt%. Subsequently, after passing through the water-absorption trap, the water content was monitored at various intervals with different gas volumes. Even after the accumulated gas volumes reached 100 L (approximately 111.1 hours of electrolysis), the water content in the DME was only $2100.9 \pm 7.4 \text{ ppm}$ (**Figure R5i**, with all raw data in **Figure R5j**), corresponding to approximately 0.003 wt% water content in the cathode chamber exhaust gas. This is significantly lower (0.0041 times) than observed without the color-changing silica gel water-adsorption trap, indicating minimal impact on product purity.

Furthermore, considering the highest water absorption capacity of color-changing silica gel (30 wt%), the required amount of silica gel for producing 1 tonne of ethylene would be:

$$m_{silica \text{ gel}} = \frac{10^6 \times 0.003\%}{0.3}$$

$$= 100 \text{ g}$$

Importantly, color-changing silica gel can be easily regenerated through drying. Note that

this direct adsorption method necessitates no additional energy input, thus having no impact on the overall energy consumption of the indirect route. In response to the reviewer's comment, the reported data on ethylene purity has been corrected. Detailed descriptions of the method for correcting the ethylene purity, considering the influence of water vapor, are presented in the revised supporting information.

Figure R5. (a) Setup for qualitative analysis of water carried out by product stream. (b) The time needed for color-changing at different gas flow rates. (c) The total charge for Br-EO reduction at various time intervals over the AC-Ag electrode, and (d) the corresponding photo for the color-changing silica gel. (e) Setup for quantitative analysis of water content carried out by product stream. (f) The content of water in DME for pristine ultra-dry DME, Electrolysis + DME, and 15 sccm + DME, and (g) the corresponding reading in Karl-Fischer titrator. (h) Setup for removing water carried out by product stream. (i) The content of water in DME collected at different gas throughputs under 15 sccm using the setup shown in Figure R5h. (j) The reading in Karl-Fischer titrator for Electrolysis + Silica gel, 15 sccm + Silica gel-0L, 15 sccm + Silica gel-22.5L, 15 sccm + Silica gel-100L.